# Riemannian classification of single-trial surface EEG and sources during checkerboard and navigational images in humans

Cédric Simar[1]*, Robin Petit[1,2], Nichita Bozga[1], Axelle Leroy[3], Ana-Maria Cebolla[3], Mathieu Petieau[3], Gianluca Bontempi[1], Guy Cheron[3,4]

**1** Machine Learning Group, Computer Science Department, Faculty of Sciences, Université Libre de Bruxelles (ULB), Brussels, Belgium, **2** Interuniversity Institute of Bioinformatics in Brussels, Université Libre de Bruxelles- Vrije Universiteit Brussel, Brussels, Belgium, **3** Laboratory of Neurophysiology and Movement Biomechanics, Neuroscience Institute, Université Libre de Bruxelles (ULB), Brussels, Belgium, **4** Laboratory of Electrophysiology, Université de Mons-Hainaut, Mons, Belgium

* cedric.simar@ulb.be

**Data Availability Statement:** The anonymized dataset necessary to replicate the study's findings reported in this article, was made publicly available

## Abstract

### Objective

Different visual stimuli are classically used for triggering visual evoked potentials comprising well-defined components linked to the content of the displayed image. These evoked components result from the average of ongoing EEG signals in which additive and oscillatory mechanisms contribute to the component morphology. The evoked related potentials often resulted from a mixed situation (power variation and phase-locking) making basic and clinical interpretations difficult. Besides, the grand average methodology produced artificial constructs that do not reflect individual peculiarities. This motivated new approaches based on single-trial analysis as recently used in the brain-computer interface field.

### Approach

We hypothesize that EEG signals may include specific information about the visual features of the displayed image and that such distinctive traits can be identified by state-of-the-art classification algorithms based on Riemannian geometry. The same classification algorithms are also applied to the dipole sources estimated by sLORETA.

### Main results and significance

We show that our classification pipeline can effectively discriminate between the display of different visual items (Checkerboard versus 3D navigational image) in single EEG trials throughout multiple subjects. The present methodology reaches a single-trial classification accuracy of about 84% and 93% for inter-subject and intra-subject classification respectively using surface EEG. Interestingly, we note that the classification algorithms trained on sLORETA sources estimation fail to generalize among multiple subjects (63%), which may be due to either the average head model used by sLORETA or the subsequent spatial filtering

without restriction at https://doi.org/10.6084/m9.figshare.14997735.v1. The dataset consists of covariance matrices of shape 12 X 12, estimated on the output of the xDAWN filtering algorithm applied to either EEG signals or mean cortical sources activations of each trial of each subject. The Python source files implementing the classification pipelines and generating the figures published in this article were also made publicly available without restriction at https://github.com/cedricsimar/TC-EEG-RSTA.

**Funding:** The authors thank the fund Leibu and the Brain & Society foundation for their support as well as T. D'Angelo, M. Dufief, E. Hortmanns, E. Pecoraro, and E. Toussaint for expert technical assistance.

**Competing interests:** The authors have declared that no competing interests exist.

failing to extract discriminative information, but reach an intra-subject classification accuracy of 82%.

## 1. Introduction

Since its discovery [1, 2], EEG has increasingly been used in fundamental, clinical, and industrial researches. For each of these domains, specific tools were successively developed. These tools include (i) the intracerebral recording with microelectrodes [3, 4] which allowed the recognition of the neuronal origin of EEG signals and a better understanding of the physiological mechanisms that underlie EEG activity; (ii) the grand averaging method, consisting in the average of a series of trials [5] triggered by a repetitive event (visual, auditory, somesthesic. . .), which opened the evoked-related potential (ERP) field studies, more recently enriched by EEG dynamics tools [6, 7] including EEG source generators [8–10]; and (iii) the use of EEG for neurofeedback and brain-computer interfaces (BCI) [11, 12]. In the past, these domains and their related tools evolved separately but the increasing accessibility of computational resources and experimental data motivated the development of transversal approaches and methodological bridges.

Visual evoked potentials (VEP) are a particular type of ERP, extracted from EEG signals recorded over the occipital cortex, that may be triggered by the display of different sorts of visual stimuli, from simple (e.g. checkerboard) [13, p., 14, 15] to more complex ones (e.g. human face, 3D or moving image) [14, 16–20]. VEP are obtained by computing the grand average of numerous trials of ongoing EEG signals (see Eq 1), resulting in well-designed and easily recognizable potentials that are subsequently used to better understand the successive processing stages of visual inputs. However, these evoked responses result from at least two different mechanisms derived from the additive or the oscillation model [8, 21–24]. For the additive model, the evoked responses result from sequential bottom-up processing of sensory inputs. This produces a specific sequence of monophasic evoked component peaks that are originally embedded in spontaneous EEG background. This latter EEG activity is considered as noise and ruled out by the subsequent averaging. For the oscillatory model, the evoked potential might result from phase-locking of the ongoing EEG rhythms within specific frequency bands. This EEG phase reorganization can be measured by the inter-trials coherency (ITC), as a response to the external stimulation. On the fundamental level, this measure is interesting only when there are no simultaneous variations (increase or decrease) of the related EEG powers. In this case, we are in presence of a pure phase-locking and the evoked responses are only due to a reorganization of the ongoing EEG oscillation. For example, this is the case of the N30 component of the somatosensory evoked potential for which 70% of amplitude is due to pure phase locking [25]. The fact that, in the majority of ERP studies, a mixed situation (power variation and phase-locking) occurs, makes basic and clinical interpretations difficult. Another disadvantage originates from the fact that, in the majority of evoked potentials studies, a grand average among a cohort of subjects is performed. Although the grand average method allows appropriate statistics [26] and practical conclusions about basic or clinical outcomes, it masks the individual peculiarities which may be critical from a clinical point of view. This problem is particularly crucial when diagnosis tools are based on grand average evoked potentials [27]. Similarly, the application of inverse modeling [10, 28] applied to grand average data provides very efficient recognition of the ERP generators [19, 29–31] but does not facilitate the determination of individual characteristics. In the face of these shortcomings of

classical ERP analysis, the introduction of machine learning tools based on different classification pipelines such as Riemannian geometry may allow better exploitation of single trials and individual characteristics in the evoked potentials domain [32–41]. The relevance of considering single-trial analyses with neuroimaging data is also discussed in [42].

We hypothesize that the aforementioned ERP components of the EEG signals contain discriminative information characterizing the visual features of the image that can be identified, in single trials, by a state-of-the-art classification pipeline based on the canonical Riemannian geometry of covariance matrices. The main contribution of this study with respect to the state-of-the-art is the proposition of a methodological framework demonstrating that it is possible to gain insights into the classical evoked potentials fields by the application of recent BCI classification techniques [36], allowing the discrimination of visual evoked responses in a single-trial approach across multiple subjects, as opposed to the classical grand average approach based solely on mean EEG signals. For this, we use a classification pipeline based on xDAWN spatial filtering [33] and Riemannian geometry applied to single-trial EEG data recorded during a visual stimulation. Riemannian geometry classifiers have received growing attention in the last few years [43], particularly due to their first-class performance in international Brain-Computer Interface (BCI) competitions [44]. Besides, special attention is given to the potential advantage of introducing inverse modeling to the Riemannian classification pipeline. Furthermore, in this study, we compare the discriminative power of our framework when trained on each subject separately (hereunder referred to as intra-subject classification) or on all subjects indistinctively (hereunder referred to as inter-subject classification). Yet, the latter is paramount to estimate how the framework classification model can generalize to unseen subjects.

We show that our present methodology can effectively discriminate between single-trial EEG signals from different visual items presentation (Checkerboard versus 3D navigational image) with a classification accuracy of about 84% and 93% for the inter-subject and intra-subject condition respectively. These successful results motivate us to introduce the "Riemannian Single-Trial Analysis" (RSTA) approach, which interest will be further discussed in comparison to the Grand Average Analysis approach commonly used in neuroscience. Additionally, a comparative ERP and RSTA analysis on the grey images displayed between visual stimuli was performed and presented as S1 File in order to show that these images, typically considered neutral, actually carry discriminative information about the preceding visual stimulus presented to the subject.

## 2. Materials and methods

### 2.1 Participants

The data was collected from 15 healthy volunteers. All participants were right-handed, had no neurological condition, and normal vision, including 3D vision. Each participant gave informed consent to the experimental procedures. All experimental protocols were approved by the Ethic comity of Université Libre de Bruxelles, CHU Brugmann, and conducted in conformity with the European Union directive 2001/20/EC of the European Parliament.

### 2.2 Experimental paradigms

Participants were watching the EGA screen of an IBM Laptop (screen of 22.0 cm height, 30.3 cm width; refresh rate of 75 Hz, resolution of 800 x 600 pixels) centered on the line of gaze at a distance of 30 cm from the eyes through a cylindrical tunnel adapted to remove any external visual interferences, as previously used by our group [19]. We presented a sequence of checkerboards comprising 96 images intermixed with 96 grey images and a sequence of 3D-Tunnel presentations containing 192 images from four randomized corridor orientations (up, down, right, and left) giving an implicit illusion of virtual navigation intermixed with grey images

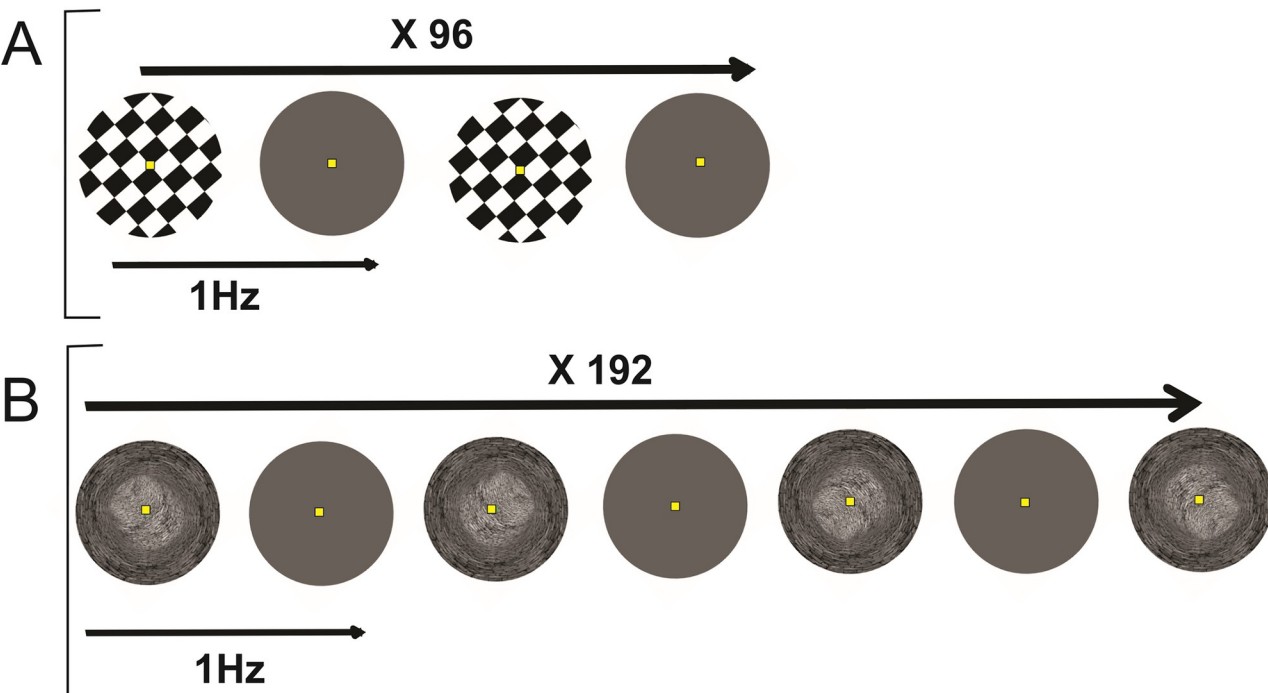

**Fig 1. (A, B)** Overview of the two different stimulation paradigms. A: paradigm 1, Checkerboard images (upper line) followed by grey screens. B: paradigm 2, 3D-Tunnels randomly presented in the 4 orientations (up, down, left, right), followed each time by a grey screen.

(Fig 1). There was no break between recording sessions of Checkerboard and 3D-Tunnel patterns. The first recording session, either displaying Checkerboard or 3D-Tunnel patterns, was random and alternated between subjects.

### 2.3 Stimulation and recording parameters

All participants performed passive observation of the aforementioned visual stimuli. Within one recording run, either 96 Checkerboard patterns or 192 3D-Tunnels were presented to one subject, alternately with a uniform grey image (Fig 1A and 1B). An identical stimulation rate (1.0 Hz) was used in both conditions. Checkerboard stimulus consisted of black and white rectangles (4.5 x 4.0 cm) alternating 96 times with the grey page, which corresponded to full-contrast black and white squares (black field 15 lx; white field 101 lx).

The grey page luminance was about 43 lx. The 3D-Tunnel was non-stereoscopic but included perspective cues generated by the OpenGL graphic libraries [45] (Fig 1A and 1B). It represented a tunnel with stone-textured walls (stone dimension 1.25 cm$^2$ at the periphery to 0.15 cm$^2$ close to the center) in the form of a pipe with a constant circular cross-section. These different stimuli with a pattern contrast of about 50% display subtended 7˚(w) × 5˚ (h) at the eye. Thus, both foveal and parafoveal retinal fields were stimulated. The luminance of the 3D-Tunnel evolved from 39 lx at the periphery to 74 lx close to the center. The presentation of each visual item (presentation time of 500 ms) was immediately followed by the presentation of a uniform grey image (also for 500 ms).

### 2.4 EEG data treatment

EEG data were recorded with an active-shield cap using 128 Ag / AgCl sintered ring electrodes and shielded co-axial cables (5–10 electrode system placements) comfortably adjusted to the

participant's head. All recordings were referenced to the left earlobe electrode. Vertical and horizontal eye movements (EOG) were recorded bipolarly. All electrode impedances were maintained below 5kΩ. Scalp potentials were amplified by ANT DC-amplifiers (ANT neuro system, the Netherlands) and digitized at a rate of 2.048 Hz using a resolution of 16 bit (range 11 mV). A band-pass filter from DC to 256 Hz and a notch filter (47.5–52.5 Hz) were also applied. Participants were asked to avoid eye blinks and to focus on the green dot presented in the middle of the screen to reduce eye artifacts. In order to verify the effectiveness of the eye fixation requirement, the number of eye movements was recorded throughout the different visual stimulation periods. For this, saccades including small saccades of about 0.8° and other eye movements were automatically selected by a Matlab (MathWorks Inc) script using eye velocity threshold. This selection was then verified by visual inspection. For all subjects, the fixation requirement was respected. Only 0.19 ± 0.09 saccades per second were recorded, irrespective of the type of image presented to the participant. Off-line data treatment and statistics were performed using EEGLAB software [6]. Artifactual portions of the EEG data were rejected after appropriate independent component analysis (ICA). Eventually, a zero-phase IIR bandpass filter with cut-off frequencies at 1 Hz and 45 Hz was applied and epochs containing samples from 0 ms before and 500 ms after the stimulus were extracted, which encompasses most of the information of the early visual potential components [16]. The averaged signal value of the pre-stimulus interval from -500 ms to 0 ms was divided from the epochs for baseline correction. More details about EEG data processing can be found in our previous study [19].

Besides, in order to facilitate future applications in clinical routines, only the following 12 electrodes uniformly distributed on the scalp have been used for classification: F3, Fz, F4, C3, Cz, C4, P3, Pz, P4, O1, Oz, and O2.

As the 3D-Tunnel can be presented in 4 different orientations, and as a decent amount of trials was required for each of them [31], the dataset originally presented an unbalanced number of Checkerboard and 3D-Tunnel trials. Thus, in order to avoid any bias in the classification results, for each subject separately, we applied a randomized undersampling, i.e. randomly removing trials from the majority class (3D-Tunnel), to obtain an equal number of Checkerboard and 3D-Tunnel trials.

## 2.5 XDAWN filtering and covariances

As described in Barachant (2014) [46], the xDAWN algorithm estimates a chosen number of spatial filters that enhance the signal-to-noise ratio of the evoked potentials for each class.

Let $E \in \mathbb{N}$ denote the number of electrodes, $X_i \in \mathbb{R}^{E \times T}$ denote the trial of index $i$ with $1 \leq i \leq N^{(k)}$, $T$ the number of time samples, and $N^{(k)}$ the number of trials from class $k$. Let $P^{(k)} \in \mathbb{R}^{E \times T}$ denote the grand average of class $k$, or equivalently the VEP of the visual stimulus corresponding to class $k$, defined as:

$$P^{(k)} = \frac{1}{N^{(k)}} \sum_{i=0}^{N^{(k)}} X_i \tag{1}$$

Let $X$ denote the matrix representing the entire signal, obtained by concatenating all the trials from the two classes.

Each spatial filter is a vector $w \in \mathbb{R}^{E \times 1}$ and is estimated to increase the signal-to-noise ratio of its related class. In other words, for class $k$ we have:

$$w^{*(k)} = \underset{w}{\operatorname{argmax}} \frac{w^\top P^{(k)} P^{(k)^\top} w}{w^\top X X^\top w} \tag{2}$$

These filters can be found by maximizing a generalized Rayleigh quotient, i.e. by solving the generalized eigenvalue problem on the matrices $P^{(k)}P^{(k)\top}$ and $XX^{\mathrm{T}}$. Only the filters associated with the $F$ highest eigenvalues are selected ($F$ being the parameterizable number of xDAWN spatial filters).

Let $W^{(k)} \in \mathbb{R}^{E \times F}$ denote the selected spatial filters for class $k$, and $W = [W^{(0)}, W^{(1)}] \in \mathbb{R}^{E \times 2F}$ the aggregation of the spatial filters. The spatially filtered data is then defined by $Z_i \in \mathbb{R}^{E \times 2F}$:

$$Z_i = W^\top X_i \tag{3}$$

We define a new matrix $\tilde{Z}_i \in \mathbb{R}^{4F \times \mathrm{T}}$ by concatenating the filtered averaged trials from both classes (0 and 1) with the spatially filtered data as follows:

$$\tilde{Z}_i = \begin{bmatrix} W^{(0)\top} P^{(0)} \\ W^{(1)\top} P^{(1)} \\ Z_i \end{bmatrix} \tag{4}$$

Finally, we can estimate the covariance matrix $\Sigma_i \in \mathbb{R}^{4F \times 4F}$ of $\tilde{Z}_i$:

$$\Sigma_i = \frac{1}{N} \tilde{Z}_i \tilde{Z}_i^T \tag{5}$$

In particular, the shape of covariance matrices estimated after filtering the data is $4F \times 4F$. In order to main covariance matrices of shape $12 \times 12$, we fixed the number of filters $F = 3$.

In practice, $\Sigma_i$ is computed using a well-conditioned estimator such as Ledoit-Wolf [47] or OAS [48].

## 2.6 Tangent space mapping

In all our classification pipelines, we apply a tangent space mapping operator to the previously estimated covariance matrices and use the mapped result as the input to a logistic regression classifier.

Covariance matrices are symmetric positive definite (SPD) and therefore do not lie in a vector space but in a convex cone [49] which has a Riemannian manifold structure, i.e. for each point of the manifold, there is an associated tangent space where a dot product is defined. In particular, we consider the tangent space at the point $\Sigma_{\mathrm{ref}}$ which corresponds to the geometric mean of the whole set of covariance matrices. More specifically, $\Sigma_{\mathrm{ref}}$ is the point minimizing the average Fréchet distance to the set of covariance matrices. The choice of $\Sigma_{\mathrm{ref}}$ is motivated by the observation from Tuzel et al., [50] that the geometric mean is the point where the mapping onto the tangent space leads to the best local approximation of the manifold.

Tangent space mapping is the action of projecting the SPD matrices from the manifold onto the associated tangent space. In the tangent space, the $n \times n$ covariance matrices will be represented by a vector of dimension $n(n + 1)/2$. This projection operator at the reference point $\Sigma_{\mathrm{ref}}$ is defined by Barachant et al., [51] as the upper triangular matrix of $\sqrt{2} \left(1_E\, 1_E^\top - I_E\right) \Phi(\Sigma)$ where $\Phi(\Sigma)$ is defined as:

$$\Phi(\Sigma) = \log_{\Sigma_{\mathrm{ref}}}(\Sigma) = \Sigma_{\mathrm{ref}}^{1/2} \log\left(\Sigma_{\mathrm{ref}}^{-1/2}\, \Sigma\, \Sigma_{\mathrm{ref}}^{-1/2}\right) \Sigma_{\mathrm{ref}}^{1/2} \tag{6}$$

where $\log_{\Sigma_{\mathrm{ref}}}(\Sigma)$ denotes the matrix logarithm of $\Sigma$ with respect to $\Sigma_{\mathrm{ref}}$.

## 2.7 Source analysis

The objective of source analysis is to improve spatial resolution, increase SNR, and detect subcortical activities not directly observable on scalp EEG. In this work, we estimated cortical and subcortical sources using the non-parametric inverse method Standardized Low Resolution Electromagnetic Tomography (sLORETA) [9, 28]. This work used the sLORETA estimation implemented in the MNE Python library [52] and which is formally defined as follows. Let $\hat{J}(\alpha) \in \mathbb{R}^{3M \times T}$ be the source estimation matrix:

$$\hat{J}(\alpha) = T(\alpha)X \coloneqq K^{\top}(KK^{\top} + \alpha I)^{-1}X,$$

where $E \in \mathbb{N}$ is the number of electrodes at the scalp surface, $M \in \mathbb{N}$ is the number of sources within the brain volume, $X \in \mathbb{R}^{E \times T}$ is the EEG signal matrix, $K \in \mathbb{R}^{E \times 3M}$ is the lead field matrix and $\alpha > 0$ is a regularization parameter.

Consider $\Sigma\big(\hat{J}(\alpha)\big)$, the covariance matrix of $\hat{J}(\alpha)$ defined as:

$$\Sigma\big(\hat{J}(\alpha)\big) = \Sigma(T(\alpha)X) = T(\alpha)\Sigma(X)T(\alpha)^{\top} = K^{\top}(KK^{\top} + \alpha I)^{-1}K.$$

$\Sigma\big(\hat{J}(\alpha)\big)$ is a $3M \times 3M$ square matrix with M diagonal blocks of shape $3 \times 3$ denoted by $\Sigma_1, \ldots, \Sigma_M$. The normalized sLORETA source estimation of the voxel $\ell$ is therefore given by:

$$\hat{J}_{\text{sLORETA}}(\alpha)_{\ell} \coloneqq \Sigma_{\ell}^{-1/2}\,\hat{J}(\alpha)_{\ell} = \Sigma_{\ell}^{-1/2}\,T(\alpha)_{\ell}\,X \in \mathbb{R}^{3 \times T}$$

where $T(\alpha)_{\ell}$ is the $\ell^{\text{th}}$ $3 \times E$ row block of $T(\alpha)$.

## 2.8 Classification pipelines

This section describes the classification pipelines illustrated in Fig 2. The accuracy of each inter-subject pipeline was estimated using a Leave-One-Subject-Out Cross-Validation (LOSO-CV) where each subject out of the 15 has been used for the validation of the classification algorithm trained using the remaining 14 subjects. Similarly, the accuracy of each intra-subject pipeline was estimated using a 4-Fold Cross-Validation where, for each subject, a quarter of trials were used for validation of the classification algorithm using the remaining three quarters. xDAWN spatial filters were estimated on the training set and applied to the validation set.

The classification pipeline of both inter-subject and intra-subject EEG signals from the 12 electrodes of 3D-Tunnel versus Checkerboard visual stimuli without inverse modeling includes (i) xDAWN spatial filtering operating on the EEG signals from all 12 electrodes, (ii) the estimation of covariance matrices, (iii) the projection onto the tangent space (TS) and (iv) a logistic regression (LR) classifier.

The classification pipeline of both inter-subject and intra-subject EEG signals of 3D-Tunnel versus Checkerboard with inverse modeling includes (i) the estimation of cortical sources based on EEG signals using the Desikan-Killiany atlas [53, 54], (ii) the averaging of cortical sources activations per atlas region, (iii) xDAWN spatial filtering operating on the 75 averaged

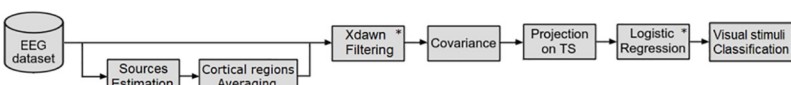

**Fig 2. Flow chart of the classification pipeline with and without inverse modeling.** An asterisk in the upper-right corner denotes that the pipeline step is supervised.

sources activations, (iv) the estimation of covariance matrices, (v) the projection onto the tangent space (TS) and (vi) a logistic regression (LR) classifier.

The results of these classification pipelines were reported using the following metrics: ROC curves, Area Under the ROC Curve (AUC), Precision-Recall curves, Average Precision (AP), confusion matrices, and Matthews correlation coefficient (MCC).

The main Python libraries used for the implementation of these pipelines are MNE [52, 55], NumPy [56], SciPy [57], and scikit-learn [58].

## 3. Results

### 3.1 ERP analysis

Fig 3 illustrates the grand average ERP differences observed between the different visual stimulations, highlighting the main characteristics of the early evoked components at around 100 ms (P100) for the Checkerboard (thick blue line) or the 3D-Tunnel (thick red line) presentation. As previously observed by our group [19], the P100 component evoked by the Checkerboard was of higher amplitude than the P100 evoked by the 3D-Tunnel, which presented a biphasic configuration during the time of the monophasic classical P100 related to the

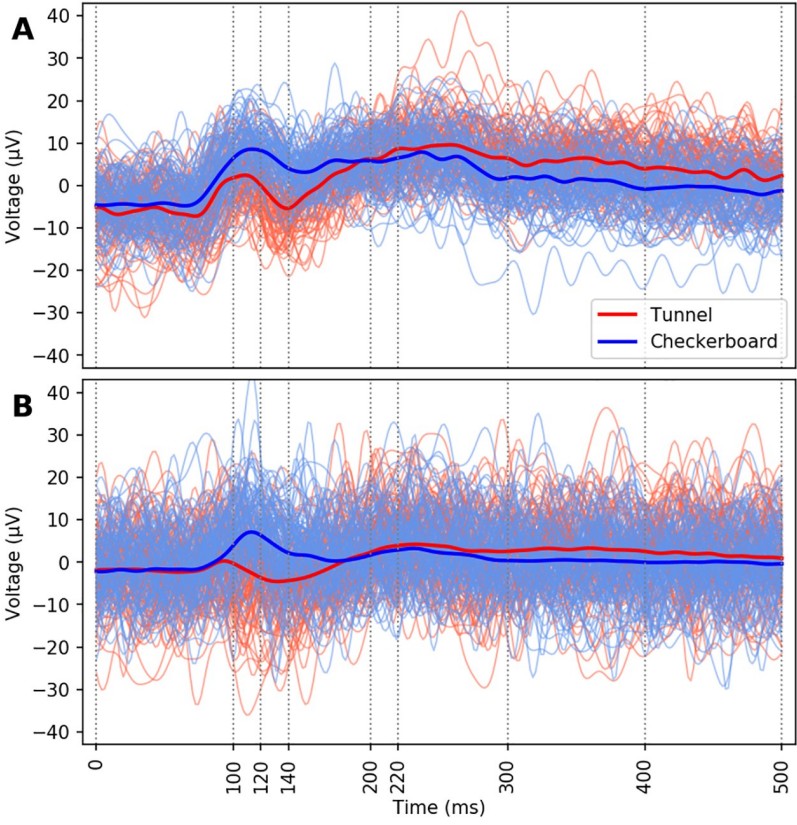

**Fig 3.** (**A**) Superimposition of EEG signals recorded on the occipital area (electrode Oz) from all the single trials of Tunnels and Checkerboards stimuli of one representative subject on which the grand average signals corresponding to the Tunnel (red lines, n = 192) and the Checkerboard (blue lines, n = 96) stimuli are superimposed. (**B**) Superimposition of EEG signals recorded on electrode Oz from 100 randomly selected trials of Tunnel stimuli and 100 randomly selected trials of Checkerboard stimuli throughout all the subjects on which the grand average signals of all trials and all subjects corresponding to the Tunnel stimuli (red lines, n = 2880 (192 trials x 15 subjects)) and the Checkerboard (blue lines, n = 1440 (96 trials x 15 subjects)) are superimposed.

Checkerboard. These earlier evoked components were followed by a P220 in the EEG signals of 3D-Tunnels and Checkerboards. The grand average signal of each class k refers to the row corresponding to the Oz electrode of the matrix $P^{(k)}$ as defined in Eq 1.

When confronted with trials coming from a single subject, differences can be visible by human visual inspection. For example, in Fig 3A we can observe differences between the single trials of Tunnels and Checkerboards between 100 and 200 ms, with the Tunnels having a noticeably lower amplitude in the selected time limits. In contrast, the discrimination by visual inspection between the two stimuli was not possible anymore when randomly chosen single trials originating from all subjects were superimposed (Fig 3B). This figure illustrates the fact that the discrimination was not possible by human visual inspection and is well representative of the difficulty of the classification task. Therefore, we may expect higher accuracies from a classifier trained and evaluated with intra-subject EEG signals compared to inter-subject EEG signals.

### 3.2 Classification performance

**3.2.1 The effect of the number and position of the electrodes.** Prior to the main classification task, we first determined if sufficient discriminative information existed with regards to the number and position of the 12 arbitrarily selected electrodes. The number of electrodes was empirically validated by quantifying their impact on the classification accuracy using a forward feature selection method. For this, we computed the distribution of classification accuracy with respect to the number of electrodes (Fig 4A) using a LOSO Cross-Validation. As a result, Fig 4A illustrates that the classification score for the Tunnel vs. Checkerboard reaches a global maximum using 9 electrodes. Considering that the positions of the 12 electrodes are uniformly distributed on the scalp and that the cross-validated classification accuracy reached a plateau before the 12th electrode, we can reasonably conclude that adding EEG signals from more electrodes will not substantially increase the amount of non-artifactual information.

Furthermore, in order to test which scalp region contains the most discriminative information, we used a triad of electrodes corresponding to the occipital, parietal, central, and frontal regions on which the classification pipeline was applied. Fig 4B illustrates this result for the Tunnel vs Checkerboard classification. The classification pipeline trained on the single trials of all subjects indistinctively reaches the best score for the occipital electrodes (75%), followed by the parietal electrodes (64%), and followed by the central and frontal electrodes (respectively 56% and 59%).

**3.2.2 Classification without inverse modeling.** The main results of the classifications pipelines were illustrated by the ROC curve (along with the AUC), the Precision-Recall curve (along with the Average Precision), and the confusion matrix. Fig 5 illustrates these performance metrics computed on the results from the inter-subject and intra-subject classification pipelines without inverse modeling. The ROC curve for the intra-subject discrimination is more arched than the curve for the inter-subject discrimination (Fig 5A), which is confirmed by the higher AUC of the first (0.98) compared to the latter (0.92). These results were further validated by the Precision-Recall curves (Fig 5B). Besides, the confusion matrices (Fig 5C) show a balanced recognition accuracy for Tunnels and Checkerboards, although, as expected, both accuracies are significantly higher for the intra-subject classification (Fig 5C right) than the inter-subject classification (Fig 5C left). The inter-subject and intra-subject classifiers without inverse modeling reached an accuracy of 84% and 93% respectively, as well as a Matthews correlation coefficient (MCC) of 0.668 and 0.860 respectively.

**3.2.3 Classification with inverse modeling.** When working with inverse modeling on the discrimination of Tunnel vs Checkerboard, the inter-subject and intra-subject classification

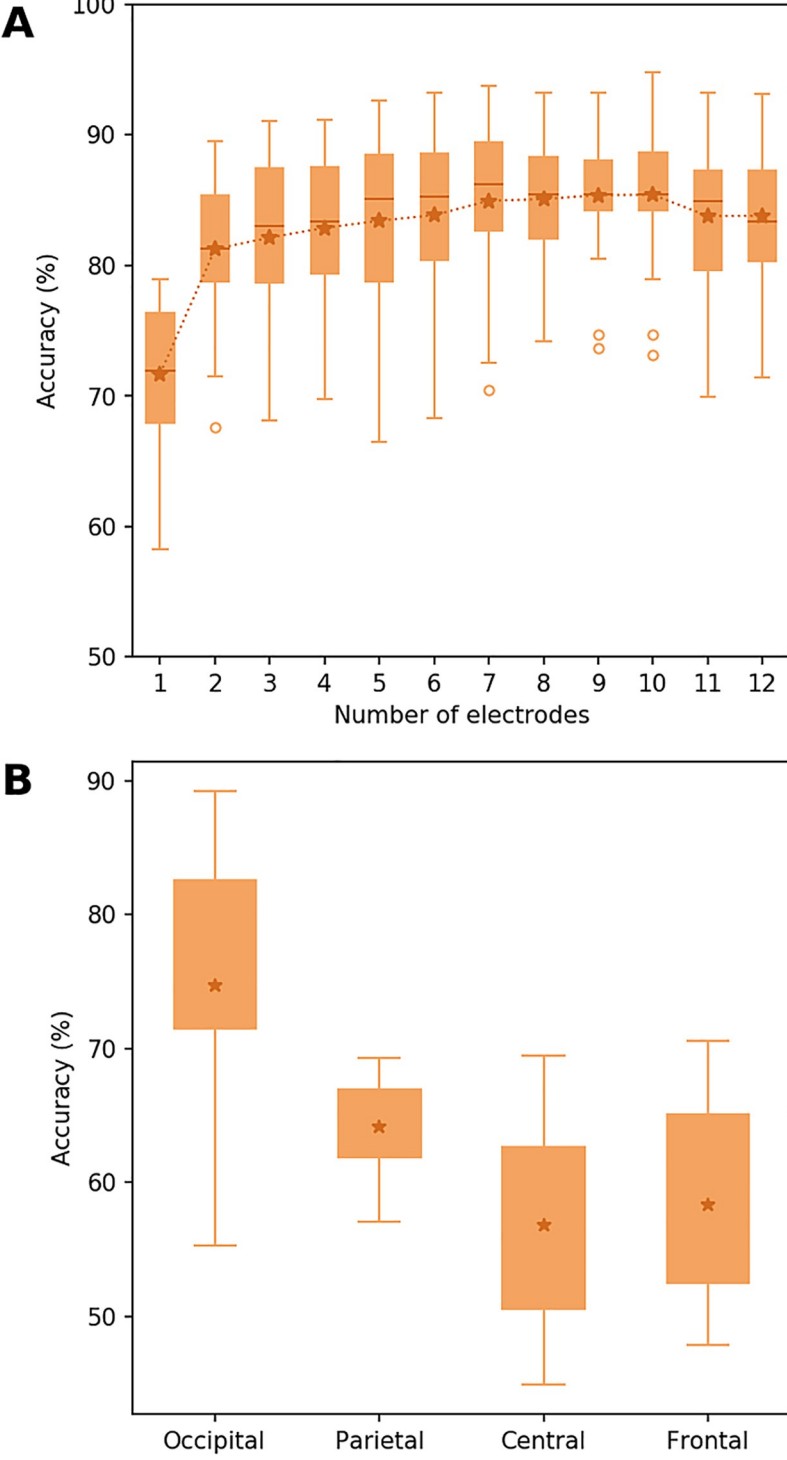

**Fig 4. Distribution of the Tunnel vs Checkerboard inter-subject cross-validated classification accuracy without inverse modeling with respect to the number of electrodes (A) and by scalp region (B) using only 3 electrodes from each locus (O1, O2, and Oz for the occipital locus, P3, P4, and Pz for the parietal locus, C3, C4, and Cz for the central locus, F3, F4 and Fz for the frontal locus).**

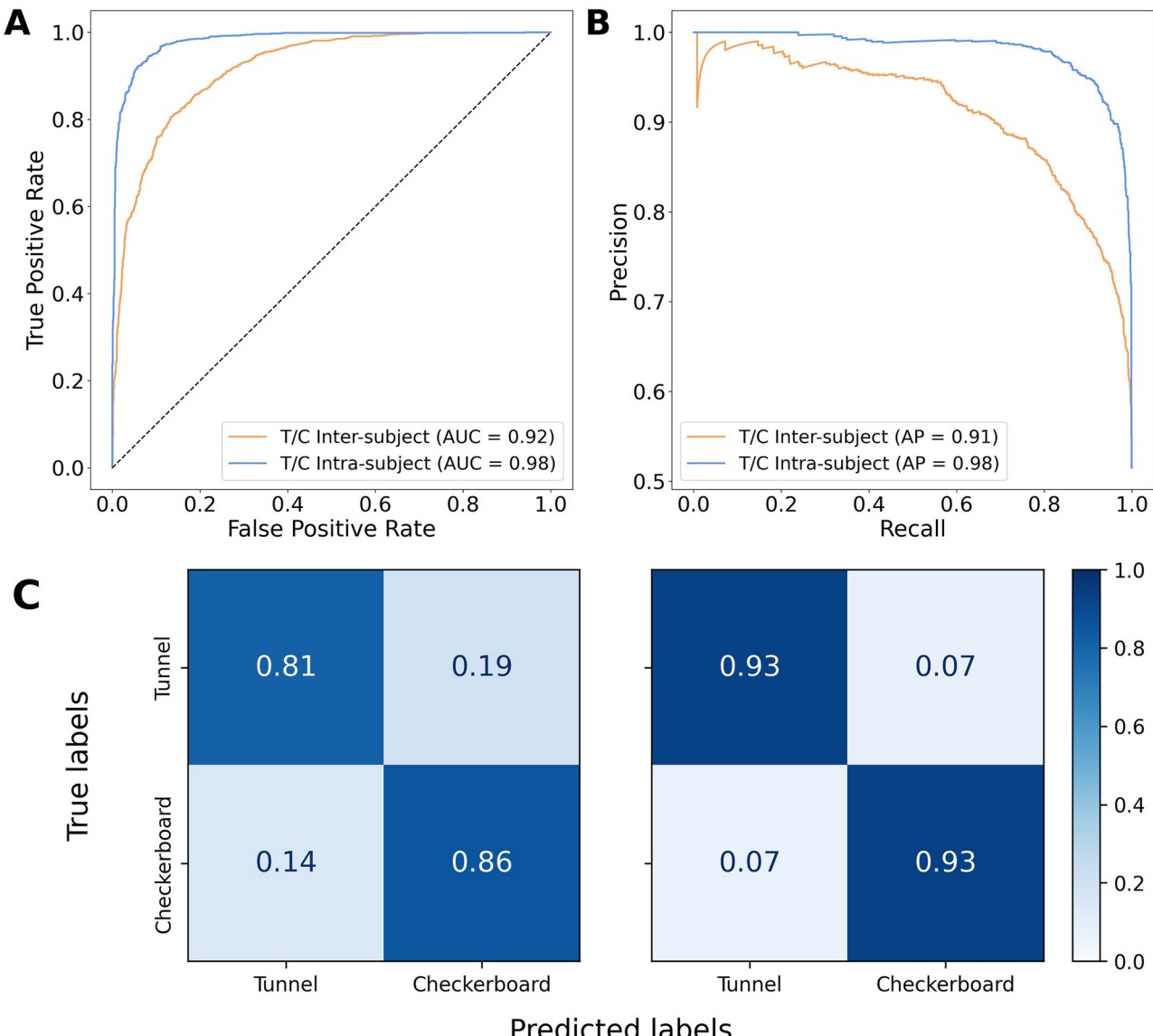

**Fig 5.** ROC (**A**) and Precision-Recall (**B**) curves for the intra-subject and inter-subject classification of Tunnel vs Checkerboard without inverse modeling. (**C**) Confusion matrices for the inter-subject (left) and intra-subject (right) classification of Tunnel vs Checkerboard without inverse modeling.

pipelines performed very differently on the same dataset. When applied to inter-subject data, the classification pipeline reached an accuracy of 63%. However, the intra-subject approach has yielded significantly better results, reaching an accuracy of 82%. Similarly to section 3.2.2, the major differences in accuracy between the two classification tasks are illustrated in Fig 6 using ROC curves and their respective AUC (Fig 6A), Precision-Recall curves (Fig 6B) as well as confusion matrices (Fig 6C). Additionally, the inter-subject and intra-subject classifiers with inverse modeling reached a MCC of 0.26 and 0.63 respectively.

We further explored the intra-subject classification by exploring the sources activations of a representative subject around 140 and 220 ms. As illustrated in Fig 7A and 7B, sources activations seem to be higher during the visualization of the Tunnels than that of the Checkerboards.

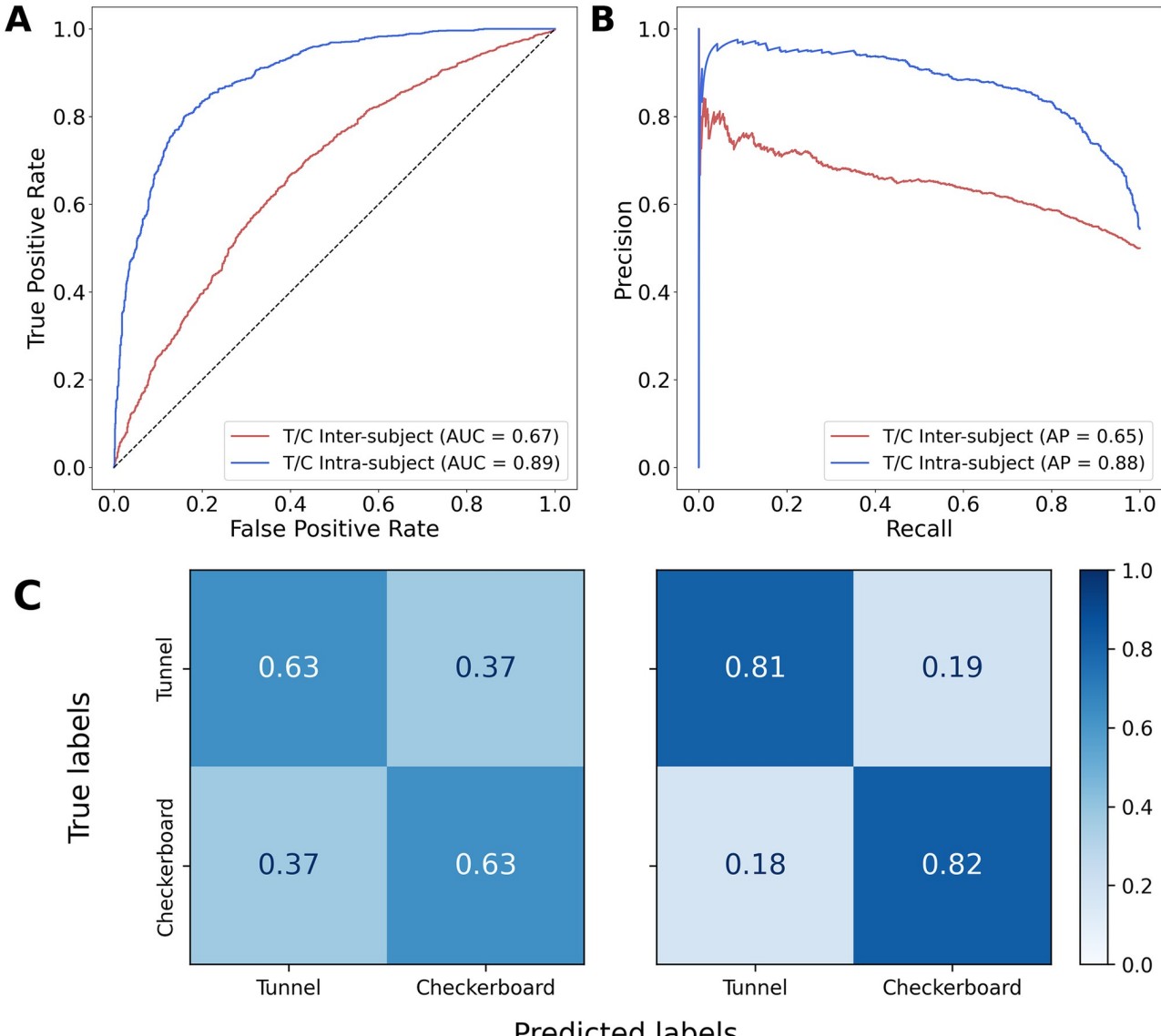

**Fig 6.** ROC (**A**) and Precision-Recall (**B**) curves for the intra-subject and inter-subject classification of Tunnel vs Checkerboard with inverse modeling. (**C**) Confusion matrices for the inter-subject (left) and intra-subject (right) classification of Tunnel vs Checkerboard with inverse modeling.

For the Tunnel, two main zones were highlighted (yellow), one in the occipital cortex and the other in the sensory-motor areas extending towards the frontal cortex. For the Checkerboard, the occipital region was also identified as well as the frontal cortex but less intensely. Although some subjects exhibited a less contrasting pattern as illustrated in Fig 7D and 7E, the differences remain in favor of higher cortical contribution for the Tunnel (occipital and frontal zones identified) while only the occipital region was identified for the Checkerboard. Fig 7C and 7F represents the activations of labeled cortical source areas as defined in the Desikan-Killiany atlas corresponding to the 3D brain visualizations from Fig 7A, 7B, 7D and 7E respectively.

**3.2.4 Summary of results.** The boxplots in Fig 8 summarizes the comparative analysis of the classification pipelines with respect to the intra-subject or inter-subject conditions. The

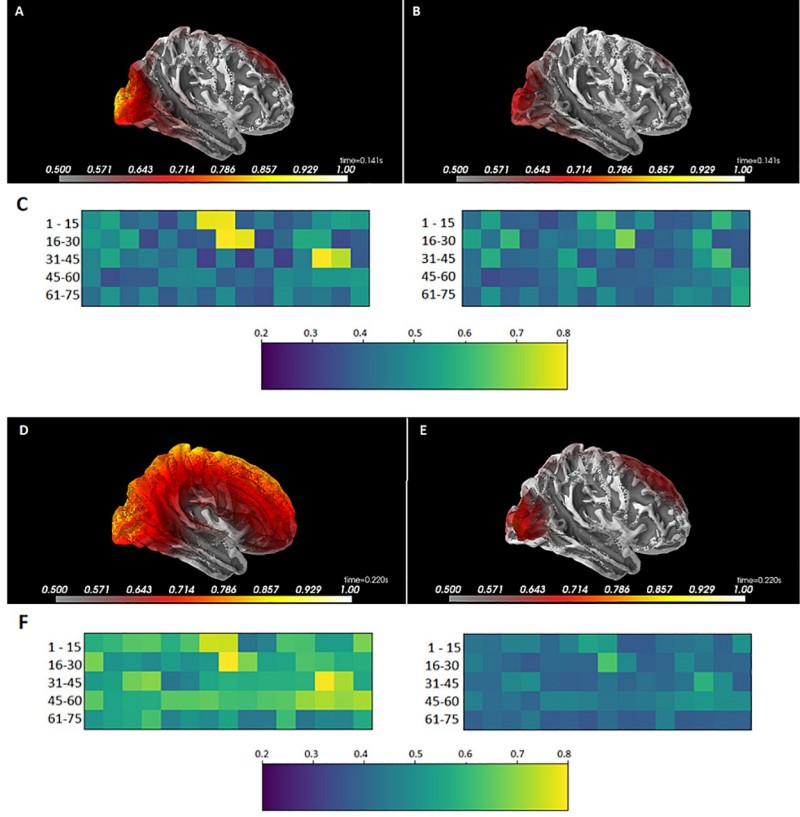

**Fig 7. Lateral view of the mean sources activations normalized estimations (A, B, D, and E) and the corresponding matrix representation of the atlas regions (C and F) labeled as defined in the Desikan-Killiany atlas at two different times in a single representative subject; 140 ms (A and B) and 220 ms (D and E) after the stimuli, during the visualization of Tunnels (A and D) and Checkerboards (B and E).** Sources activations are averaged from 10 milliseconds before to 10 milliseconds after the N140 (140 ms) and the P220 (220 ms) respectively.

classification results for the intra-subject and inter-subject classification pipelines were computed using a 4-fold cross-validation and a Leave-One-Subject-Out Cross-Validation respectively. We observed that the accuracy for each classification task was significantly higher for the intra-subject compared to the inter-subject condition (p < 0.001). Additionally, we observed a statistical difference between the classification accuracy of the Tunnel vs Checkerboard with and without inverse modeling in the inter-subject condition (p < 0.001) as well as in the intra-subject condition (p < 0.001). These statistical comparisons were computed using a Mann–Whitney U test.

**3.2.5 Comparison of classification pipelines.** As suggested by a reviewer, after the comparative analysis in section 2.8, we compare here the classification accuracy of our approach with other classification pipelines using the same cross-validation methodology. The purposes of this comparison are to (i) quantify the benefits of applying xDAWN filtering and projecting covariance matrices onto the tangent space, and (ii) establish the robustness and generalizability of our approach with respect to other classification pipelines.

To that end, Fig 9 compares the mean cross-validated accuracy, with respect to the intra-subject or inter-subject conditions, computed on the following classification pipelines with and without inverse modelling:

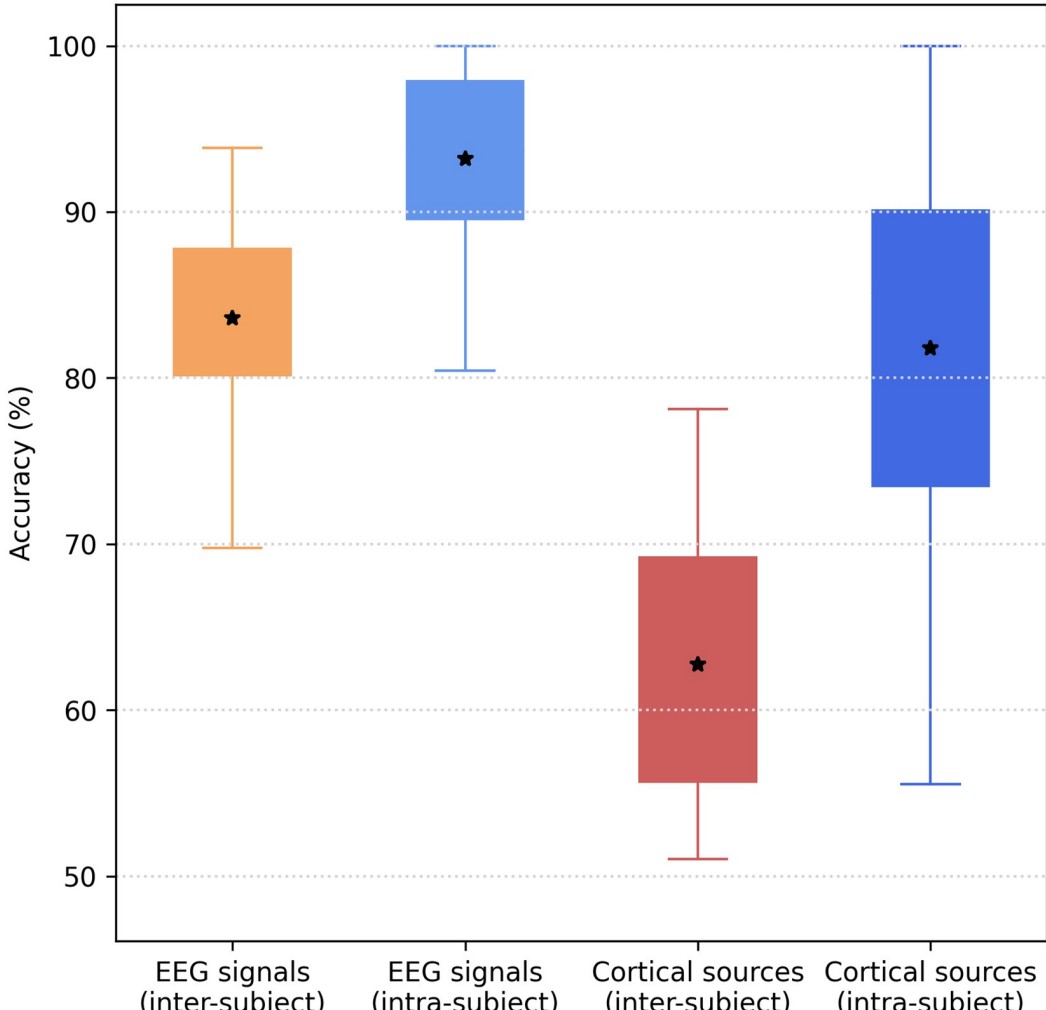

**Fig 8. Summary of the accuracies reached by the inter-subject and intra-subject classification pipelines based on EEG signals and (sub)cortical sources of Tunnel and Checkerboard visual stimuli.**

- *LR on vectorized EEG signals/sources activations*: the EEG signals or cortical sources activations are unrolled into a 1-dimensional vector and classified using a logistic regression (LR).

- *Covariance matrices and LR*: covariance matrices of EEG signals or cortical sources are estimated, unrolled into a 1-dimensional vector and classified using a logistic regression.

- *Covariance matrices, TS and LR*: covariance matrices of EEG signals or cortical sources are estimated, projected onto the tangent space and classified using a logistic regression.

- *xDAWN covariance and LR*: covariance matrices are estimated from the result of xDAWN spatial filtering applied on EEG signals or cortical sources, unrolled into a 1-dimensional vector and classified using a logistic regression.

- *CSP and LR*: the output of the Common Spatial Patterns (CSP) algorithm [59, 60] applied on EEG signals or cortical sources is classified using a logistic regression.

| Classification pipeline | Mean accuracy | |
|---|---|---|
| **EEG signals** | **Intra-subject** | **Inter-subject** |
| LR on vectorized EEG signals | 0,49 | 0,67 |
| Covariance matrices and LR | 0,47 | 0,51 |
| Covariance matrices, TS and LR | 0,89 | 0,55 |
| xDAWN covariance and LR | 0,48 | 0,52 |
| CSP and LR | 0,86 | 0,56 |
| xDAWN covariance, TS and LR * | **0,93** | **0,84** |
| **Cortical sources** | | |
| LR on vectorized sources activations | 0,57 | 0,58 |
| Covariance matrices and LR | 0,66 | 0,56 |
| Covariance matrices, TS and LR | 0,81 | 0,53 |
| xDAWN covariance and LR | 0,53 | 0,57 |
| CSP and LR | 0,77 | 0,51 |
| xDAWN covariance, TS and LR * | **0,82** | **0,63** |

**Fig 9. Comparison of the mean accuracies reached by different inter-subject and intra-subject classification pipelines based on EEG signals and (sub)cortical sources.** The asterisk designates the classification pipeline used in our approach.

- *xDAWN covariance, TS and LR*: covariance matrices are estimated from the result of xDAWN spatial filtering applied on EEG signals or cortical sources, projected onto the tangent space and classified using a logistic regression.

The comparison of classification results from these different pipelines illustrated in Fig 9 highlights several interesting patterns. Firstly, we can observe that the classification pipeline used in our approach outperforms the other pipelines in both intra-subject and inter-subject conditions, using either EEG signals or cortical sources. Secondly, we observe that the use of xDAWN spatial filtering significantly improves the classification accuracy in the inter-subject condition ($p < 0.001$), i.e. inter-subject generalizability, but not in the intra-subject condition ($p > 0.001$). Whereas, conversely, the use of the projection onto the tangent space significantly improves the classification accuracy in the intra-subject condition ($p < 0.001$) but not in the inter-subject condition ($p > 0.001$), although this latter absence of accuracy improvement in the inter-subject condition may also be explained by the difficulty to discriminate inter-subject EEG signals and cortical sources without xDAWN spatial filtering for this task. Lastly, we observe that the classification pipeline used in our approach significantly outperforms the CSP algorithm in the inter-subject condition ($p < 0.001$) but not in the intra-subject condition ($p > 0.001$). These statistical comparisons were computed using a Mann–Whitney U test.

## 4. Discussion

In this work, we leveraged the BCI methodology and related mathematical tools in order to better analyze the EEG signals commonly summarized and interpreted by the analysis of the evoked potentials components. Since the renewing of the evoked potentials interpretation following the oscillation models by Makeig et al., [9] and the introduction of EEGLab software [6], the dynamic aspects of the event-related potentials (ERP) contributed to better access to the underlying neurophysiological mechanisms sustaining the ERP components. At the same time, the development of the BCI domain [12, 61] necessitated online procedures in order to extract pertinent neural information from noisy environments. As BCI procedures are specifically devoted to work with only one subject at a time, the BCI tools are well appropriate to

decipher individual EEG-evoked responses and to extend our understanding beyond grand average representations of the ERP.

We show that it is possible to effectively discriminate between the display of different visual items (Checkerboard versus 3D navigational image) in single related EEG signals corresponding to one subject. Besides, the present methodology allows us to demonstrate that the single-trial discrimination based on Riemannian classification pipelines can be generalized to multiple subjects constituting a single batch of single trials. In other words, we may here introduce the "Riemannian Single-Trial Analysis" (RSTA) approach, as opposed to the Grand Average Analysis (GAA) approach commonly used in clinical neuroscience. Although the latter is useful for highlighting statistically significant differences between subject populations, GAA is prone to capture artificial constructs due to individual peculiarities. Indeed, it is possible that averaged evoked components may be sign inverted or phase jittered depending on the subjects included in the grand average. Therefore, these statistically significant differences may be spurious and may not represent the underlying physiological characteristics of the observed evoked potential resulting from the GAA. Besides, a statistical difference induced by the GAA may not necessarily imply physiological plausibility nor genuine discriminative property that should be the basis of a clinical application. As opposed to the GAA, the objective of the RSTA approach is to allow, using interpretability techniques on Riemannian classification pipelines, the identification and extraction of subject-, task-, and trial-specific underlying discriminative patterns that can unlock a deeper understanding of the fundamental neurophysiological mechanisms.

It was shown in previous studies [62–66] that the single-trial configuration was able to discriminate cognitive evoked components, allowing the possibility to detect the intermittency of cognitive events during repetitive stimulation. Although the evoked components are not visually distinguishable in the present single-trial configuration, as displayed in Fig 3B, we show that our classification pipeline based on covariance matrices and Riemannian geometry is able to discriminate between the presentation of the two displays, reaching a single-trial classification accuracy of about 84% and 93% for inter-subject and intra-subject classification respectively.

Despite the methodological difference about the paradigm and the type of evoked potentials or brain signal recordings, the results of the present classification pipelines based on Riemannian geometry corroborate the single-trial classification performances of similar Riemannian approaches applied on objects familiarity recognition [67], motor imagery [68, 69], P300 BCI [70] and EEG respiratory states [71].

Using inverse modeling based on swLORETA [10], we previously showed that, whatever the presented image (Checkerboard or Tunnel), the same generators of the P100 were located in the occipital cortex (BA18 and BA19) and the right inferior temporal cortex (BA20). However, the left fusiform cortex (BA37) was additionally recruited only for the Checkerboard but not for the 3D-Tunnel. In the present study, in spite of the fact that a limited number of electrodes (n = 12) was used, inverse modeling based on sLORETA mainly reproduces these results using the Desikan-Killiany atlas [53, 54] but the inter-subject classification pipeline based on sLORETA sources estimations reaches significantly lower accuracy when using estimated sources (63%) rather than EEG signals (84%). In contrast, the same classification method applied to intra-subject EEG reaches an accuracy and an AUC of 82% and 0.92 respectively. This performance is comparable to the intra-subject classification performances based on EEG signals (93%). The lower performance of inter-subject classification using sLORETA sources estimations may result from either (i) the morphological differences between the subject-specific cortical dipoles orientation and the average head model used to compute the sLORETA lead field matrix that may have blurred the signal differentiation between the two images or (ii) the subsequent xDAWN spatial filtering not being able to extract discriminative

information that may be located in different, subject-specific, cortical atlas regions. Conversely, this problem does not occur when only one subject is considered. This indicates in some way the limit of the sensitivity of the present method. However, the present study shows that such Riemannian classification pipelines allow, on the one hand, scalp EEG signal discrimination throughout multiple subjects which could be used in an object visual recognition BCI and, on the other hand, EEG neural generators discrimination in a single subject which could be used in a clinical context. This latter aspect is in line with the present trend to favor the decoding of the individual brain activity [72, 73] which outperforms the use of the grand average from multiple subjects.

### 4.1 Limitations

There are several limitations to the current study. Firstly, the limited number of electrodes (n = 12) used, to favor clinical and BCI applications, may have reduced the precision of the 3D inverse modeling estimation performed with sLORETA and may thus have impacted the related classification performance. Secondly, although the number of subjects (n = 15) was limited compared to other EEG studies, the dataset, consisting of the number of trials per subject times the number of subjects, was considered sufficient to obtain meaningful inter-subject classification results using the LOSO-cross-validated RSTA approach. Nevertheless, further studies with a larger number of participants will be necessary in order to generalize to population. Thirdly, we are also aware that the presented results are directed towards the global aspect of the visual image, though the two different images presented different physical characteristics (bottom-up features such as luminescence and contrast) and different representational contents (top-down influence). The distinctive influence of these physical and neuro-cognitive factors on the Riemannian classification pipelines will be further investigated. Future research could also be oriented towards new stimulation paradigms in order to differentiate EEG signals evoked by images with similar physical characteristics but different representational contents. Besides, in order to better understand the fundamental underlying neurophysiological mechanisms, future research may also be dedicated to quantifying the importance of the discriminative patterns identified by the RSTA approach.

## 5. Conclusion

This work shows that classification pipelines based on Riemannian geometry can effectively discriminate between the display of different visual items (Checkerboard versus 3D navigational image) in single EEG trials. The presented methodology reaches a single-trial classification accuracy of about 84% and 93% for inter-subject and intra-subject classification respectively using surface EEG, which was shown to be significantly better than the accuracy of other state-of-the-art pipelines. Furthermore, the classification algorithms trained on sLORETA estimation fail to generalize among multiple subjects (63% accuracy) but reach an intra-subject classification accuracy of 82% which allows future functional links between neuro-imagery and EEG dynamics discrimination. In this context, the development of new advanced mathematical frameworks will be paramount to develop a deeper understanding of the underlying communicational dynamics inside the neural network topography upon which the Riemannian classification pipeline performances emerged.

## Supporting information

**S1 File.**
(DOCX)

## Acknowledgments

The authors thank the fund Leibu and the Brain & Society foundation for their support as well as T. D'Angelo, M. Dufief, E. Hortmanns, E. Pecoraro, and E. Toussaint for expert technical assistance.

## Author Contributions

**Conceptualization:** Axelle Leroy, Ana-Maria Cebolla, Guy Cheron.

**Data curation:** Cédric Simar, Robin Petit, Nichita Bozga, Axelle Leroy, Ana-Maria Cebolla, Mathieu Petieau, Gianluca Bontempi, Guy Cheron.

**Formal analysis:** Cédric Simar, Robin Petit, Nichita Bozga, Ana-Maria Cebolla, Gianluca Bontempi, Guy Cheron.

**Funding acquisition:** Axelle Leroy.

**Investigation:** Gianluca Bontempi.

**Methodology:** Cédric Simar, Robin Petit, Axelle Leroy, Ana-Maria Cebolla, Gianluca Bontempi, Guy Cheron.

**Resources:** Mathieu Petieau, Guy Cheron.

**Software:** Cédric Simar, Mathieu Petieau.

**Supervision:** Cédric Simar, Ana-Maria Cebolla, Gianluca Bontempi, Guy Cheron.

**Validation:** Cédric Simar, Robin Petit, Guy Cheron.

**Visualization:** Robin Petit.

**Writing – original draft:** Cédric Simar, Robin Petit, Nichita Bozga, Ana-Maria Cebolla, Gianluca Bontempi, Guy Cheron.

**Writing – review & editing:** Cédric Simar, Robin Petit, Ana-Maria Cebolla, Guy Cheron.

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
