## [Decision Letter · Decision Letter 0]

22 Mar 2021

PONE-D-21-04037

Riemannian classification of single-trial surface EEG and sources during checkerboard and navigational images in humans

PLOS ONE

Dear Dr. Simar,

Thank you for submitting your manuscript to PLOS ONE. After careful consideration, we feel that it has merit but does not fully meet PLOS ONE’s publication criteria as it currently stands. Therefore, we invite you to submit a revised version of the manuscript that addresses the points raised during the review process.

Specifically:

While the reviewers pointed out that the results reported in the present paper are relevant and promising and that the proposed methodological framework is robust, they suggested major revisions in the description of the experiment and in the presentation of the results.

In particular, the data underlying the results are unavailable - not conforming to PLos Policy - and one reviewer postponed his/her complete review untill the data/method parameters are provided. Two other reviewers added specific comments (see below).

We look forward to receiving your revised manuscript.

Kind regards,

Yury Ivanenko

Academic Editor

PLOS ONE

Journal Requirements:

2a) If there are ethical or legal restrictions on sharing a de-identified data set, please explain them in detail (e.g., data contain potentially sensitive information, data are owned by a third-party organization, etc.) and who has imposed them (e.g., an ethics committee). Please also provide contact information for a data access committee, ethics committee, or other institutional body to which data requests may be sent.

2b) If there are no restrictions, please upload the minimal anonymized data set necessary to replicate your study findings as either Supporting Information files or to a stable, public repository and provide us with the relevant URLs, DOIs, or accession numbers. For a list of acceptable repositories, please see http://journals.plos.org/plosone/s/data-availability#loc-recommended-repositories.

Reviewers' comments:

Reviewer's Responses to Questions

**Comments to the Author**

1. Is the manuscript technically sound, and do the data support the conclusions?

Reviewer #1: Yes

Reviewer #2: Partly

Reviewer #3: Partly

2. Has the statistical analysis been performed appropriately and rigorously? 

Reviewer #1: Yes

Reviewer #2: Yes

Reviewer #3: No

3. Have the authors made all data underlying the findings in their manuscript fully available?

Reviewer #1: No

Reviewer #2: No

Reviewer #3: Yes

4. Is the manuscript presented in an intelligible fashion and written in standard English?

Reviewer #1: Yes

Reviewer #2: No

Reviewer #3: Yes

5. Review Comments to the Author

Reviewer #1: The data underlying the results are unavailable - not conforming to PLos Policy. The raw data are likely protected/confidential, which is fine. The covariance matrices which are the basis for the tangent space projection and classification presented can however be shared without chance to breach confidentiality. The code underlying the analysis is to be declared/available. A method paper without these is no use to anyone.

All data/method parameters should be disclosed fully following eg. https://cobidasmeeg.wordpress.com/.

Until those criteria are met, independently of the quality of the work (which looks good), publication is not acceptable IMO. Once satisfied, I'll provide a complete review.

Reviewer #2: The Authors Cédric Simar et al. proposed a new approach to classify ERP evoked by different visual stimuli, at single trial level. To do so, they did identify a classification pipeline based on the Riemannian geometry of covariance matrices applied to EEG scalp signals and reconstructed source signals (sLORETA algorithm employed for the purpose). It is certainly not a novel topic considering the wide attention that it has always garnered, especially in the field of BCI applications, but the results reported in the present paper are relevant and promising. Although the proposed methodological framework is in my opinion robust, a more precise description is needed in order to make those results reproducible and, more importantly, applicable to several other applications different from visual stimulation. The impact of this paper would benefit from the possible generalization of the classifier performance in other contexts. Although I do not think that further analyses are necessary, major revisions in the description of the experiment and in the presentation of the results need to be done before publishing the paper. In the present shape, I found it confusing and almost not informative as a consequence. I suggest the Authors stressing the hypothesis behind this work and even more the impact. In the following, I listed more specific comments.

1. In the Introduction section there is a general lack of information and connection between the different parts of this study and it does not provide a comprehensive description of the state of the art able to clearly define the space in which this paper will make a difference. The employment of this method for BCI applications is not even mentioned so it is difficult to understand the importance of moving from the GA approach, used successfully for decades. Another example is the fact that the Authors mentioned the results relative to the classification accuracy for the inter- and intra-subject condition without first introducing those concepts and the reason why both those analysis were performed. Since the objective of the paper is to classify at single trial level, it is not obvious to understand how inter-subjects data should be involved in the training phase and what the hypothesis behind this choice is.

2. Why did the Authors design the two visual tasks with unbalanced number of trials (96 trials for the Checkerboard task and more than double for the 3D-Tunnel)? My guess is that this choice was dictated by the fact that the 3D-Tunnel can be presented in 4 different orientations and a decent amount of trials is required for each of them in order to test possible differences between them. Please include the rationale behind this choice in the paragraph 2.2.

3. Paragraph 2.3 – The sentence “Checkerboard pattern and the 3D-Tunnel were alternatively presented with a uniform grey image (Figure 1A, B)” is confusing in addition to poorly written. The adverb “alternately”, for example, is more appropriate than “alternatively”. The figure certainly helps but reading this description it is not clear how the stimuli are actually presented. It suggests that Checkerboard pattern and 3D-Tunnel are presented within the same run along with gray screens, which is not correct. Please, rephrase.

4. The paragraph 2.4 is called “EEG data treatment and event related potentials”. Please replace event related potentials with ERP for consistency with the rest of the paper. More important is the fact that ERP are the title but not even mentioned in the text. Please, provide some details about the ERP analysis maybe justifying the choice of the windows of interest, which are the ERP the Authors are interested in and they expect to see in each experimental condition with specific latencies. The sentence at the beginning of paragraph 3.1: “As previously observed by our group [7], the P100 component evoked by the Checkerboard was of higher amplitude than the P100 evoked by the 3D Tunnel, which presented a biphasic configuration during the time of the monophasic classical P100 related to the Checkerboard”, is a good example and should be moved either in the Introduction or in the Method section.

5. Still in the same paragraph (2.4), the Authors provided a list of the 12 EEG channels used for the ERP analysis and subsequential classification. However, most of the results obtained at scalp level are presented for Oz only. This becomes clear only reading the caption of the reported figures. Please, include this information both in the Method section and in the Result section and provide an explanation for this choice, even simply specifying whether it is because ERP results were similar for all the sensors or, vice versa, because Oz was gave the only meaningful results (mentioning the analysis presented in the Paragraph 3.2.1 that should also be described in the method section instead of the results section directly).

6. Left earlobe electrode as EEG reference is not a popular choice for scalp data acquisition. Please justify.

7. In the description of the source analysis, the Authors did not specify the number of scalp sensors (probably all of them) and reconstructed sources. The fact that only 12 of them are included in the ERP analysis might lead to the wrong conclusion that those signals are also enough to estimate accurately the brain activity in cortical sites. Important aspects of the analysis like the chosen atlas and the number of sources of interest is discussed only in the paragraph 2.8 - which should instead focus on the classification pipeline - making the methodological framework confusing. Other important information, such as whether they considered the reconstructed signal on the centroid of each region of interest or the average of all dipoles, or even the number of voxels used in the lead-field are also missing. The Authors mentioned “cortical regions average” but this step is not clearly described anywhere. Those details are necessary to reproduce the results without using the MNE software chosen by the Authors.

8. The Authors wrote that measures like ROC curves, Precision-Recall curves and confusion matrices are appropriate since the dataset is not balanced (paragraph 2.8). Can they please explain more this relationship? I think that those are the best measures to evaluate the performances of a classifier regardless of the number of trials difference. In this case, that sentence might be inaccurate.

9. I strongly suggest revising Paragraph 3.1, including the captions of Figure 4 and 5. The results description is not clear and very long sentences end up not carrying any message. Here few examples: i) “visual distinction” is a poor choice when the main result here is the distinction between 2 visual stimuli. I would replace that expression with “The discrimination by visual inspection between the 2 stimuli, was not possible anymore if/when…” ii) what’s a “single raw EEG signal”? Does it mean single trial? iii) Panels A and B in Figure 5 are redundant. My suggestion is either to report a plot similar to Figure 4 (it would be the most consistent way to present the results besides making immediate the comparison between the two conditions) or to keep only one of the two current panels.

10. Figure 1: please improve the quality/resolution of the 3D-TUNNEL image.

11. Paragraph 3.2.1: the title of this section is not precise because most of the following text refers to the validation of the number of electrodes. I suggest something like: “The effect of the electrodes number and position”. I also would appreciate some clarifications: considering that no statistical analysis has been performed and reported in Figure 6A, how would the Authors justify that (for example) the condition 9 electrodes is better than 10? Which qualitative measures did they observe? Also, it is not clear to me how the classifier was trained. Is it “on the single trials of all subjects” in both cases? Should this evaluation be performed for inter- and intra-subjects classification separately to be sure that the best number and position of the electrodes is the same?

12. In the same vein of my previous comment, did the Authors considered performing a similar analysis after the source reconstruction? If I understand correctly, the final classifier evaluation without inverse modeling, has been performed using 3 occipital electrodes. How did the Authors use the data from 75 cortical sources? Did they consider their optimal number and location? If not, why did they non consider it an interesting aspect of the analysis? Reducing the number of ROIs might be a key aspect of this pipeline and it would reduce the computational time. Something about this topic has been done by the Authors and reported in the second half of Paragraph 3.2.4 but I could not understand in which way these results are related with the classification analysis and if they are just a mere observation of the brain activations. I want to conclude this comment saying that the importance and relevance of the presented results are completely hidden by the lack of details (both in the main text and in the figures caption), by the quality of the text and by the poor organization of the paragraphs. I strongly suggest the Authors to focus on improving the presentation of the results.

13. Figure 7/8/9: For the sake of clarity and consistency between the results obtained before and after the source reconstruction, I suggest reporting figures containing the same information. If the comparison between inter- and intra-subjects coded by red and pink lines respectively have been chosen for the classification with inverse modeling, similar figures should be reported for the classification without inverse modeling. I suggest combining figure 7 and 8 removing the results relative to the gray screen (eventually the Authors can report them separately in the supplementary). The alternative would be to report the results for the gray screens classification obtained in the source space.

14. The discussion section needs to be improved on the same aspects I listed so far for the other sections with an additional lack of continuity between the different parts. The discussion opens with several comments very specifically BCI-related when this concept was barely touched in the introduction and in the abstract. I suggest i) revising the text because not well written and difficult to follow; ii) mention that contributing to improve the use of ERP in BCI is one of the main goals of the paper toward the end of the Introduction. As I pointed out several times, there is a general lack of common thread throughout the whole paper and highlighting the purpose and which results play the most important role in achieving this goal might help the reader to orient among the different sections.

15. The Authors should acknowledge somewhere the limitation of the study. They can decide to dedicate a separate paragraph of the paper or to add some comments at the end of the discussion. The main points I suggest mentioning are the limited number of subjects (particularly meaningful because we are talking about healthy subjects performing an easy task and because they are used to validate a new approach for the first time) and the fact that the results need to be taken with caution because only tested on two visual stimuli under extremely controlled conditions (no peripheral view). The Authors do not have to forget the final purpose of the classification at single trial level, which is the ability to use measures like ERP in real time in neurofeedback or BCI applications, particularly meaningful in the rehabilitation field where the signals might be acquired in less ideal conditions without significantly degrading the performance of the classifier. If the Authors are thinking making available the code for their pipeline, this should be mentioned because more scientists could apply it and help validating/improving the results obtained on these 15 subjects.

16. Please double check for typos and inaccuracies. Here few examples:

a. The introduction opens with a confusing sentence in which the words “evoked potentials” appear twice. I suggest deleting “which generate the well-defined components of the evoked potentials” that might be redundant.

b. Introduction: “In THE PRESENT STUDY, we use a classification pipeline based on xDAWN spatial filtering [23] and Riemannian geometry applied TO SINGLE-TRIAL EEG DATA recorded during A visual stimulation”.

c. Figure 8C: please add a label to specify that the confusion matrix on the left refers to the actual visual stimulus and the one on the left to the gray screens. Also, as a general comment, the confusion matrices are too big considering that the only relevant information is the number in the middle of each square.

d. Figure 9: The Authors never talked about “patients”. Please replace with “subjects”.

e. I suggest replacing everywhere “Raw data” with “Scalp data” or “Sensors data”. EEG data filtered and segmented are not “raw”.

Reviewer #3: The development of techniques that allow the discrimination of individual brain states, based on the non-invasive recording of brain activity, is probably a fundamental step for the extensive use of BCIs. In this sense, the present work constitutes a step forward in the right direction and therefore of great interest and importance.

The manuscript is well written and the techniques used are novel and of great interest today. The discussion about the current approach versus conventional techniques is especially interesting.

However, there are important details that need to be checked and some minor errors that need to be fixed.

Major comments

From a methodological point of view, it is not clear whether this is one or two separate experiments.

In what order were the experiments done? Was there a first session in which the checkerboard and the grey images were alternated, followed by another session in which the images of the tunnels and the grey images alternated?

Was there a break between sessions? Were the tunnel images always presented in the same order? Was the checkerboard session always the first or were they alternated between subjects? In my opinion, the description of the methodology is very poor.

Even more important to understanding the experiments is why the images were presented in this way. Maybe it would have been better to have presented the three images (checkerboard/ tunnel/gray) in a randomized way in the same session? Perhaps it would have been more appropriate to present the same number of images (checkerboard /tunnel /gray)?

The datasets appear to have come from previous experiments and have been re-analyzed to obtain the present results. This is not problematic, but if it is, and is mentioned in the manuscript, it explains some of the inconsistencies discussed here.

In another sense, it is not clear from the manuscript what value gray images have in these experiments. The fact that the gray images have less visual structure does not necessarily imply that they are "neutral images" or controls for any situation. This is especially important because the approach to the analysis of the results would be much more robust if a comparison were made between the three images instead of comparisons between only two. Making confusion matrices using the three images would give more information and strengthen the classification capacity of the technique.

Finally, the statistical analysis of the comparisons is not described and is poor. Although ROC is probably the best analysis, the use of "accuracy" independently of the rest of the elements of the confusion matrix is not entirely appropriate. A more effective measure is that based on the Matthews Correlation Coefficient, which is much more illustrative of the behavior of the matrix, especially in the case of binary classifications.

Minor comments

3.1 ERP analysis:

- Fig 3 Where the baselines were taken?

- In the description of figure 4, in the main text, figures 4 A and B are cited instead of the left side and right side of figure 4.

In the legend of figure 4 (left) it is said that the signals were recorded between 100 and 300 ms but it must be between 0 and 300 ms.

3.2.1 Validation of the electrode selection:

Results about Fig. 6. Statistical analyzes are not mentioned.

In Fig. 6a, the authors talk about global maxima using 9 electrodes, but it is not stated how it is shown that there are significant differences between the electrode configurations. On the contrary, it seems that when more than 2 electrodes were used, the accuracy did not increase significantly.

Similarly for Fig. 6b, the description of the results lacked statistical analysis.

3.2.2 Inter-subject class…

- Fig. 7 C (and 8C and 9C) the description of the color code is mixing.

. … Accuracies are significantly higher… Where are the statistics??

3.2.4 inter-subject and intra-subject …

- It is not clear in the text what is to be shown with figures 7C and 8C. Does this comparison refer to the gray images that were presented with the checkerboard and with the tunnels?

If so, what the results would be saying is that the classification system between the gray images is more related to the checkerboard and the tunnel than to the gray images themselves. This requires further explanation in the text.

- Fig. 9. Must be Intra-subjects or Inter-subjects instead intra-patients or inter-patients

6. PLOS authors have the option to publish the peer review history of their article (what does this mean?). If published, this will include your full peer review and any attached files.

Reviewer #1: **Yes: **Cyril Pernet

Reviewer #2: **Yes: **Alessandra Anzolin

Reviewer #3: No

---

## [Author Response · Author response to Decision Letter 0]

25 Jul 2021

First of all, we would like to thank the reviewers for their time, their positive comments and their suggestions that significantly improved the quality and the clarity of the revised manuscript. The response to the reviewers comments were uploaded as a pdf file named "Response to Reviewers". Nevertheless, please find hereunder the copy of the content from this file.

Reviewer #1:

The data underlying the results are unavailable - not conforming to PLos Policy. The raw data are likely protected/confidential, which is fine. The covariance matrices which are the basis for the tangent space projection and classification presented can however be shared without chance to breach confidentiality. The code underlying the analysis is to be declared/available. A method paper without these is no use to anyone.

All data/method parameters should be disclosed fully following eg. https://cobidasmeeg.wordpress.com/.

Until those criteria are met, independently of the quality of the work (which looks good), publication is not acceptable IMO. Once satisfied, I'll provide a complete review.

R: Thank you for your positive comment, conforming to PLos Policy, the anonymized dataset necessary to replicate the study’s findings reported in this article, was made publicly available without restriction at https://doi.org/10.6084/m9.figshare.14997735.v1. The dataset consists of covariance matrices of shape 12 X 12, estimated on the output of the xDAWN filtering algorithm applied to either EEG signals or mean cortical sources activations of each trial of each subject. The Python source files implementing the classification pipelines and generating the figures published in this article were also made publicly available without restriction at https://github.com/cedricsimar/TC-EEG-RSTA.

Reviewer #2:

The Authors Cédric Simar et al. proposed a new approach to classify ERP evoked by different visual stimuli, at single trial level. To do so, they did identify a classification pipeline based on the Riemannian geometry of covariance matrices applied to EEG scalp signals and reconstructed source signals (sLORETA algorithm employed for the purpose). It is certainly not a novel topic considering the wide attention that it has always garnered, especially in the field of BCI applications, but the results reported in the present paper are relevant and promising. Although the proposed methodological framework is in my opinion robust, a more precise description is needed in order to make those results reproducible and, more importantly, applicable to several other applications different from visual stimulation. The impact of this paper would benefit from the possible generalization of the classifier performance in other contexts. Although I do not think that further analyses are necessary, major revisions in the description of the experiment and in the presentation of the results need to be done before publishing the paper. In the present shape, I found it confusing and almost not informative as a consequence. I suggest the Authors stressing the hypothesis behind this work and even more the impact. In the following, I listed more specific comments.

R: Thank you for the positive comments and the suggestions to make the manuscript more didactical and clearer. We now completely revised the manuscript according to your suggestions to better highlight the advantage of the present method for a more generalized application in the BCI field. 

We really appreciated the time you took for writing such an extensive review of our work. Your suggestions really helped us to significantly improve the quality and clarity of the article. 

1. In the Introduction section there is a general lack of information and connection between the different parts of this study and it does not provide a comprehensive description of the state of the art able to clearly define the space in which this paper will make a difference. The employment of this method for BCI applications is not even mentioned so it is difficult to understand the importance of moving from the GA approach, used successfully for decades. 

R: Thank you for this important remark, we have now better introduced the general context and the motivation to present our methodological framework as opposed to the GA approach as follows:

“Since its discovery [1], [2], EEG has increasingly been used in fundamental, clinical, and industrial researches. For each of these domains, specific tools were successively developed. These tools include (i) the intracerebral recording with microelectrodes [3], [4] which allowed the recognition of the neuronal origin of EEG signals and a better understanding of the physiological mechanisms that underlie EEG activity; (ii) the grand averaging method, consisting in the average of a series of trials [5] triggered by a repetitive event (visual, auditory, somesthesic…), which opened the evoked-related potential (ERP) field studies, more recently enriched by EEG dynamics tools [6], [7] including EEG source generators [8]–[10]; and (iii) the use of EEG for neurofeedback and brain-computer interfaces (BCI) [11], [12]. In the past, these domains and their related tools evolved separately but the increasing accessibility of computational resources and experimental data motivated the development of transversal approaches and methodological bridges.

…

We hypothesize that the aforementioned ERP components of the EEG signals contain discriminative information characterizing the visual features of the image that can be identified, in single trials, by a state-of-the-art classification pipeline based on the canonical Riemannian geometry of covariance matrices. The main contribution of this study with respect to the state-of-the-art is the proposition of a methodological framework demonstrating that it is possible to gain insights into the classical evoked potentials fields by the application of recent BCI classification techniques [36], allowing the discrimination of visual evoked responses in a single-trial approach across multiple subjects, as opposed to the classical grand average approach based solely on mean EEG signals. For this, we use a classification pipeline based on xDAWN spatial filtering [33] and Riemannian geometry applied to single-trial EEG data recorded during a visual stimulation. Riemannian geometry classifiers have received growing attention in the last few years [42], particularly due to their first-class performance in international Brain-Computer Interface (BCI) competitions [43]. Besides, special attention is given to the potential advantage of introducing inverse modeling to the Riemannian classification pipeline.”

Another example is the fact that the Authors mentioned the results relative to the classification accuracy for the inter- and intra-subject condition without first introducing those concepts and the reason why both those analysis were performed. Since the objective of the paper is to classify at single trial level, it is not obvious to understand how inter-subjects data should be involved in the training phase and what the hypothesis behind this choice is.

R: Thank you for this justified remark, we now introduce these important concepts in the introduction as follows:

“Furthermore, in this study, we compare the discriminative power of our framework when trained on each subject separately (hereunder referred to as intra-subject classification) or on all subjects indistinctively (hereunder referred to as inter-subject classification). Yet, the latter is paramount to estimate how the framework classification model can generalize to unseen subjects.”

2. Why did the Authors design the two visual tasks with unbalanced number of trials (96 trials for the Checkerboard task and more than double for the 3D-Tunnel)? My guess is that this choice was dictated by the fact that the 3D-Tunnel can be presented in 4 different orientations and a decent amount of trials is required for each of them in order to test possible differences between them. Please include the rationale behind this choice in the paragraph 2.2.

R: You are right about the rationale to have an unbalanced number of trials. We now better explain the reason why we have a unbalanced number of recorded trials and the randomized undersampling performed to have a balanced dataset to train our model.

In paragraph 2.4 (EEG data treatment) we wrote:

“As the 3D-Tunnel can be presented in 4 different orientations, and as a decent amount of trials was required for each of them [31], the dataset originally presented an unbalanced number of Checkerboard and 3D-Tunnel trials. Thus, in order to avoid any bias in the classification results, for each subject separately, we applied a randomized undersampling, i.e. randomly removing trials from the majority class (3D-Tunnel), to obtain an equal number of Checkerboard and 3D-Tunnel trials.”

And in paragraph 2.8 (Classification pipelines) we introduced the metrics as follows: 

The results of these classification pipelines were reported using the following metrics: ROC curves, Area Under the ROC Curve (AUC), Precision-Recall curves, Average Precision (AP), confusion matrices, and Matthews correlation coefficient (MCC).

3. Paragraph 2.3 – The sentence “Checkerboard pattern and the 3D-Tunnel were alternatively presented with a uniform grey image (Figure 1A, B)” is confusing in addition to poorly written. The adverb “alternately”, for example, is more appropriate than “alternatively”. The figure certainly helps but reading this description it is not clear how the stimuli are actually presented. It suggests that Checkerboard pattern and 3D-Tunnel are presented within the same run along with gray screens, which is not correct. Please, rephrase.

R: Thank you for this remark, we clarified this sentence by rephrasing to “Within one recording run, either 96 Checkerboard patterns or 192 3D-Tunnels were presented to one subject, alternately with a uniform grey image (Figure 1A, B)”.

4. The paragraph 2.4 is called “EEG data treatment and event related potentials”. Please replace event related potentials with ERP for consistency with the rest of the paper. More important is the fact that ERP are the title but not even mentioned in the text. Please, provide some details about the ERP analysis maybe justifying the choice of the windows of interest, which are the ERP the Authors are interested in and they expect to see in each experimental condition with specific latencies. The sentence at the beginning of paragraph 3.1: “As previously observed by our group [7], the P100 component evoked by the Checkerboard was of higher amplitude than the P100 evoked by the 3D Tunnel, which presented a biphasic configuration during the time of the monophasic classical P100 related to the Checkerboard”, is a good example and should be moved either in the Introduction or in the Method section.

R: Thank you for you pertinent remark. The following sentences was added in the introduction to better define the ERP meaning: 

• “the grand averaging method, consisting in the average of a series of trials [5] triggered by a repetitive event (visual, auditory, somesthesic…), which opened the evoked-related potential (ERP) field studies”. 

• “Visual evoked potentials (VEP) are a particular type of ERP, extracted from EEG signals recorded over the occipital cortex, that may be triggered by the display of different sorts of visual stimuli, from simple (e.g. checkerboard) [13, p.], [14], [15] to more complex ones (e.g. human face, 3D or moving image) [14], [16]–[20]. VEP are obtained by computing the grand average of numerous trials of ongoing EEG signals (see Equation 1), resulting in well-designed and easily recognizable potentials that are subsequently used to better understand the successive processing stages of visual inputs.”

As for the signal window of interest, we took the whole window of 500ms which encompasses most of the information of the early visual potential components. 

5. Still in the same paragraph (2.4), the Authors provided a list of the 12 EEG channels used for the ERP analysis and subsequential classification. However, most of the results obtained at scalp level are presented for Oz only. This becomes clear only reading the caption of the reported figures. Please, include this information both in the Method section and in the Result section and provide an explanation for this choice, even simply specifying whether it is because ERP results were similar for all the sensors or, vice versa, because Oz was gave the only meaningful results (mentioning the analysis presented in the Paragraph 3.2.1 that should also be described in the method section instead of the results section directly).

R: Thank you for this remark. All occipital electrodes (O1, O2 and Oz) exhibited similar ERP dynamics so we chose Oz more for its central occipital position rather than a difference in the ERP quality.

6. Left earlobe electrode as EEG reference is not a popular choice for scalp data acquisition. Please justify.

R: Thank you for pointing our choice of reference which remains an important but unresolved debate, which hopefully, did not have any major consequences on the presently reported classification results. There is no best reference according to the EEGLab community, however it is very important to indicate where the reference was taken and to maintain the same reference during the whole study in order to allow the reproducibility of these results by other groups. In addition, in case of the median nerve stimulation in one side of the body (e.g. Desmedt and Cheron, 1980), the left and the right electrodes signals (farfield potentials P11, P14) measured with non-cephalic reference are not exactly the same. However, in the present bilateral visual stimulation, left and right electrodes signals should be identical when referenced with non-cephalic reference. Because we are aware of the linear mixing recorded over the scalp, we have performed an analysis of the source estimation in which the average reference is a previous requirement.

7. In the description of the source analysis, the Authors did not specify the number of scalp sensors (probably all of them) and reconstructed sources. The fact that only 12 of them are included in the ERP analysis might lead to the wrong conclusion that those signals are also enough to estimate accurately the brain activity in cortical sites. Important aspects of the analysis like the chosen atlas and the number of sources of interest is discussed only in the paragraph 2.8 - which should instead focus on the classification pipeline - making the methodological framework confusing. Other important information, such as whether they considered the reconstructed signal on the centroid of each region of interest or the average of all dipoles, or even the number of voxels used in the lead-field are also missing. The Authors mentioned “cortical regions average” but this step is not clearly described anywhere. Those details are necessary to reproduce the results without using the MNE software chosen by the Authors.

R: Thank you for this important remark. In this work, we chose to consider the same number of 12 electrodes for the pipeline with and without inverse modelling. We acknowledge that 12 electrodes are certainly not enough to estimate cortical activity with good precision. Nevertheless, we believed that, methodologically, it would be biased to compare the performances of classification pipelines trained with two different numbers of electrodes, and thus two different quantities of information. This limitation of the study was better highlighted in Section 4.1 Limitations.

Regarding the cortical regions average, this step is now described in Section 2.8 Classification pipelines as : “(i) the estimation of cortical sources based on EEG signals using the Desikan-Killiany atlas [52], [53], (ii) the averaging of cortical sources activations per atlas region, (iii) xDAWN spatial filtering operating on the 75 averaged sources activations”.

8. The Authors wrote that measures like ROC curves, Precision-Recall curves and confusion matrices are appropriate since the dataset is not balanced (paragraph 2.8). Can they please explain more this relationship? I think that those are the best measures to evaluate the performances of a classifier regardless of the number of trials difference. In this case, that sentence might be inaccurate.

R: Thank you for your question and this important remark. This sentence in the first version of our manuscript was indeed confusing and did not fully reflect the data treatment implemented. In order to clarify this point, we added the following sentenced in paragraph 2.4 : “As the 3D-Tunnel can be presented in 4 different orientations, and as a decent amount of trials was required for each of them [31], the dataset originally presented an unbalanced number of Checkerboard and 3D-Tunnel trials. Thus, in order to avoid any bias in the classification results, for each subject separately, we applied a randomized undersampling, i.e. randomly removing trials from the majority class (3D-Tunnel), to obtain an equal number of Checkerboard and 3D-Tunnel trials.” 

Concerning the reported metrics, ROC curves, Precision-Recall curves, confusion matrices and MCC are indeed appropriate to evaluate the performances of a classifier regardless of the number of trials difference. Nevertheless, we wanted to stress that, unlike simple accuracy, these measures are generally more informative to assess and compare the performance of classification models.

9. I strongly suggest revising Paragraph 3.1, including the captions of Figure 4 and 5. The results description is not clear and very long sentences end up not carrying any message. Here few examples: i) “visual distinction” is a poor choice when the main result here is the distinction between 2 visual stimuli. I would replace that expression with “The discrimination by visual inspection between the 2 stimuli, was not possible anymore if/when…” ii) what’s a “single raw EEG signal”? Does it mean single trial? iii) Panels A and B in Figure 5 are redundant. My suggestion is either to report a plot similar to Figure 4 (it would be the most consistent way to present the results besides making immediate the comparison between the two conditions) or to keep only one of the two current panels.

R: Thank you for your remarks. 

For (i), we replaced the expression in the article with your proposed one. 

For (ii), we replaced “single raw EEG signal” by “EEG signal from single trials” in the caption of Figure 4 and by “EEG signals recorded on electrode Oz from all the single trials of Tunnels and Checkerboards stimuli” in the caption of Figure 5. 

For (iii), since the analysis of grey images were moved to the supplementary materials section, we partially rewrote and condensed the ERP analysis in the revised version to make it clearer and avoid redundancy, as you suggested.

10. Figure 1: please improve the quality/resolution of the 3D-TUNNEL image.

R: Thank you for your comment, the quality of the 3D-Tunnel image (Figure 1A and 1B) was improved in the revised version of the manuscript.

11. Paragraph 3.2.1: the title of this section is not precise because most of the following text refers to the validation of the number of electrodes. I suggest something like: “The effect of the electrodes number and position”. I also would appreciate some clarifications: considering that no statistical analysis has been performed and reported in Figure 6A, how would the Authors justify that (for example) the condition 9 electrodes is better than 10? Which qualitative measures did they observe? Also, it is not clear to me how the classifier was trained. Is it “on the single trials of all subjects” in both cases? Should this evaluation be performed for inter- and intra-subjects classification separately to be sure that the best number and position of the electrodes is the same?

R: Thank you for the title suggestion, we changed this accordingly in the revised version.

Thank you for your remark concerning the number of electrodes. Even though we report in Section 3.2.1 “The effect of the number and position of the electrodes” that “the classification score for the Tunnel vs. Checkerboard reaches a global maximum using 9 electrodes”, we actually used the whole set of 12 electrodes to train the main classification algorithms and report the main results of the article. The purpose of checking the classification scores with regards to the number and position of the electrodes was to evaluate if adding more electrodes would effectively add more information that would increase the classification scores. The fact that we use the whole set of 12 electrodes or 75 mean cortical sources activations was made clearer in the revised version by adding the following sentences in Section 2.8: “xDAWN spatial filtering operating on the EEG signals from all 12 electrodes” and “xDAWN spatial filtering operating on the 75 averaged sources activations”. Moreover, we added the following paragraph in Section 3.2.1: “Considering that the positions of the 12 electrodes are uniformly distributed on the scalp and that the cross-validated classification accuracy reached a plateau before the 12th electrode, we can reasonably conclude that adding EEG signals from more electrodes will not substantially increase the amount of non-artifactual information”.

In both cases, we computed these results using only an inter-subjects classifier trained on the single trials of all subjects using a LOSO Cross-Validation. We understand your remark about the interest to check the effect of the number of electrodes on the intra-subjects but, finally, the most challenging task concerns the inter-subjects discrimination in single trials.

12. In the same vein of my previous comment, did the Authors considered performing a similar analysis after the source reconstruction? If I understand correctly, the final classifier evaluation without inverse modeling, has been performed using 3 occipital electrodes. How did the Authors use the data from 75 cortical sources? Did they consider their optimal number and location? If not, why did they non consider it an interesting aspect of the analysis? Reducing the number of ROIs might be a key aspect of this pipeline and it would reduce the computational time. Something about this topic has been done by the Authors and reported in the second half of Paragraph 3.2.4 but I could not understand in which way these results are related with the classification analysis and if they are just a mere observation of the brain activations. I want to conclude this comment saying that the importance and relevance of the presented results are completely hidden by the lack of details (both in the main text and in the figures caption), by the quality of the text and by the poor organization of the paragraphs. I strongly suggest the Authors to focus on improving the presentation of the results.

13. Figure 7/8/9: For the sake of clarity and consistency between the results obtained before and after the source reconstruction, I suggest reporting figures containing the same information. If the comparison between inter- and intra-subjects coded by red and pink lines respectively have been chosen for the classification with inverse modeling, similar figures should be reported for the classification without inverse modeling. I suggest combining figure 7 and 8 removing the results relative to the gray screen (eventually the Authors can report them separately in the supplementary). The alternative would be to report the results for the gray screens classification obtained in the source space.

R: Thank you for your comment. We did not perform the study of the effect of the electrodes number and position on the inverse modelling analysis. The final classifier evaluation without inverse modelling was based on all 12 selected electrodes. You are right that we used mean cortical sources from 75 areas that are used as direct input to the xDAWN filtering algorithm. In order to better clarify this in our manuscript we included the sentences “operating on the EEG signals from all 12 electrodes” and “operating on the 75 averaged sources activations” for the description of the classification pipelines of raw EEG and cortical sources respectively.

We well understand your idea to make a similar testing about the 75 cortical sources as we did for the 12 electrodes for the classification based raw EEG signals. However, this was not presently made because (i) the first aim of the present study was to demonstrate the feasibility to classify raw EEG signals in inter-subjects with Riemannian geometry; (ii) in the field of inverse modelling, it is well-admitted that a number of 12 electrodes is quite limited (this was recognized in paragraph 4.1 Limitations introduced in the new discussion) and that the study of the influence of the number of related sources does not represent the same scientific interest in this context. However, the present results offer the advantage to use a more practical and cheaper EEG cap as used in clinical applications and to prove that, even with this simplified configuration, the inverse modelling was able to reach substantial intra-subjects classification accuracy. The question whether the increase of scalp electrodes could contribute to a better classification based on inverse modelling should need a complete new future study. 

We followed your recommendation to simplify the second half of Paragraph 3.2.4 which consisted on a detailed description of the anatomical regions pointed out by the inverse-modelling study.

Thank you for your pertinent remark about the clarity of the results presentation. We followed your suggestion to (i) transfer the figures containing the classification results of the grey screens in the supplementary and (ii) to combine Figures 7 and 8 without the grey information. This indeed simplify and clarify the presentation of the results. 

14. The discussion section needs to be improved on the same aspects I listed so far for the other sections with an additional lack of continuity between the different parts. The discussion opens with several comments very specifically BCI-related when this concept was barely touched in the introduction and in the abstract. I suggest i) revising the text because not well written and difficult to follow; ii) mention that contributing to improve the use of ERP in BCI is one of the main goals of the paper toward the end of the Introduction. As I pointed out several times, there is a general lack of common thread throughout the whole paper and highlighting the purpose and which results play the most important role in achieving this goal might help the reader to orient among the different sections.

R: Thank you for this comment. We have made the best we can to significantly increase the fluidity and clarity of the manuscript following your recommendations in order to highlight the originality and the interest for future applications in the BCI and clinical fields.

15. The Authors should acknowledge somewhere the limitation of the study. They can decide to dedicate a separate paragraph of the paper or to add some comments at the end of the discussion. The main points I suggest mentioning are the limited number of subjects (particularly meaningful because we are talking about healthy subjects performing an easy task and because they are used to validate a new approach for the first time) and the fact that the results need to be taken with caution because only tested on two visual stimuli under extremely controlled conditions (no peripheral view). The Authors do not have to forget the final purpose of the classification at single trial level, which is the ability to use measures like ERP in real time in neurofeedback or BCI applications, particularly meaningful in the rehabilitation field where the signals might be acquired in less ideal conditions without significantly degrading the performance of the classifier. If the Authors are thinking making available the code for their pipeline, this should be mentioned because more scientists could apply it and help validating/improving the results obtained on these 15 subjects.

R: Thank you for your comment. We agree that an additional paragraph concerning the limitations of the study should be profitable for the readers. Following your suggestions, we wrote this paragraph to better indicate the present weaknesses and shortcomings, but also the future actions which could be taken to improve the present study. The following paragraph was added at the end of the discussion of the revised version: “There are several limitations to the current study. Firstly, the limited number of electrodes (n=12) used, to favor clinical and BCI applications, may have reduced the precision of the 3D inverse modeling estimation performed with sLORETA and may thus have impacted the related classification performance. Secondly, although the number of subjects (n=15) was limited compared to other EEG studies, the dataset, consisting of the number of trials per subject times the number of subjects, was considered sufficient to obtain meaningful inter-subject classification results using the LOSO-cross-validated RSTA approach. Nevertheless, further studies with a larger number of participants will be necessary in order to generalize to population. Thirdly, we are also aware that the presented results are directed towards the global aspect of the visual image, though the two different images presented different physical characteristics (bottom-up features such as luminescence and contrast) and different representational contents (top-down influence). The distinctive influence of these physical and neuro-cognitive factors on the Riemannian classification pipelines will be further investigated. Future research could also be oriented towards new stimulation paradigms in order to differentiate EEG signals evoked by images with similar physical characteristics but different representational contents. Besides, in order to better understand the fundamental underlying neurophysiological mechanisms, future research may also be dedicated to quantifying the importance of the discriminative patterns identified by the RSTA approach.”

16. Please double check for typos and inaccuracies. Here few examples:

a. The introduction opens with a confusing sentence in which the words “evoked potentials” appear twice. I suggest deleting “which generate the well-defined components of the evoked potentials” that might be redundant.

R: Thank you for your comment, we deleted the suggested sentence and double-checked for typos and inaccuracies. 

b. Introduction: “In THE PRESENT STUDY, we use a classification pipeline based on xDAWN spatial filtering [23] and Riemannian geometry applied TO SINGLE-TRIAL EEG DATA recorded during A visual stimulation”.

R: The introduction has been modified following your comment and other reviewers comments.

c. Figure 8C: please add a label to specify that the confusion matrix on the left refers to the actual visual stimulus and the one on the left to the gray screens. Also, as a general comment, the confusion matrices are too big considering that the only relevant information is the number in the middle of each square.

R: Figure 7 and 8 have been changed according to your previous requirement.

d. Figure 9: The Authors never talked about “patients”. Please replace with “subjects”.

R: Thank you, we replaced every occurrence of patients by subjects in the text and in the graphs.

e. I suggest replacing everywhere “Raw data” with “Scalp data” or “Sensors data”. EEG data filtered and segmented are not “raw”.

R: We replaced each occurrence of “raw signals” by “EEG signals”

Reviewer #3:

The development of techniques that allow the discrimination of individual brain states, based on the non-invasive recording of brain activity, is probably a fundamental step for the extensive use of BCIs. In this sense, the present work constitutes a step forward in the right direction and therefore of great interest and importance.

The manuscript is well written and the techniques used are novel and of great interest today. The discussion about the current approach versus conventional techniques is especially interesting.

However, there are important details that need to be checked and some minor errors that need to be fixed.

R: Thank you for the positive and encouraging comments about our study. Your suggestions and recommendations really helped us to significantly improve the quality and clarity of the article.

Major comments

From a methodological point of view, it is not clear whether this is one or two separate experiments.

In what order were the experiments done? Was there a first session in which the checkerboard and the grey images were alternated, followed by another session in which the images of the tunnels and the grey images alternated?

R: Thank you for this comment, this is only one experiment in which, within one recording run, either 96 Checkerboard patterns or 192 3D-Tunnels were presented to one subject, alternately with a uniform grey image (Figure 1A, B). We clarified this point in the revised version at paragraph 2.3 Stimulation and recording parameters.

Was there a break between sessions? Were the tunnel images always presented in the same order? Was the checkerboard session always the first or were they alternated between subjects? In my opinion, the description of the methodology is very poor.

R: There was no break between recording sessions of checkerboard and tunnels patterns. The direction of the tunnel were presented in a random order. The first recording session, either checkerboard or tunnel, was random and alternated between subjects. These information are now added in Section 2.2 Experimental paradigms. 

Even more important to understanding the experiments is why the images were presented in this way. Maybe it would have been better to have presented the three images (checkerboard/ tunnel/gray) in a randomized way in the same session? Perhaps it would have been more appropriate to present the same number of images (checkerboard /tunnel /gray)? The datasets appear to have come from previous experiments and have been re-analyzed to obtain the present results. This is not problematic, but if it is, and is mentioned in the manuscript, it explains some of the inconsistencies discussed here.

R: Thank you for your comment. In Section 2.4 EEG data treatment, we now better explain that the present dataset originated from a previously published study [32]. We have thus maintained the original unbalanced number of Checkerboard and 3D-Tunnel trials, which come from the fact that the 3D-Tunnel can be presented in 4 different orientations and a decent amount of trials was required for each of them. Additionally, we know explain the subsequent randomized undersampling performed to have a balanced dataset to train our model. In paragraph 2.4 (EEG data treatment) we wrote:

“As the 3D-Tunnel can be presented in 4 different orientations, and as a decent amount of trials was required for each of them [31], the dataset originally presented an unbalanced number of Checkerboard and 3D-Tunnel trials. Thus, in order to avoid any bias in the classification results, for each subject separately, we applied a randomized undersampling, i.e. randomly removing trials from the majority class (3D-Tunnel), to obtain an equal number of Checkerboard and 3D-Tunnel trials.” 

In another sense, it is not clear from the manuscript what value gray images have in these experiments. The fact that the gray images have less visual structure does not necessarily imply that they are "neutral images" or controls for any situation. This is especially important because the approach to the analysis of the results would be much more robust if a comparison were made between the three images instead of comparisons between only two. Making confusion matrices using the three images would give more information and strengthen the classification capacity of the technique.

R: Thank you for this pertinent remark about the presence of the alternated grey images. This is classically performed in visual evoked studies where different type of images are successively presented to the subjects. The grey images are used to avoid post-fusion effect between images, allowing a sort of reset between the presentation of two images. As suggested by one reviewer, the results related to the grey images classification are not central to the present aim of the study and were thus moved to a supplementary materials section. Moreover, Figures 3, 4, 5, 6 and 8 are now modified accordingly for highlighting the essential classification results between checkerboards and tunnels. Figures including results with grey screens were move to the supplementary materials section.

Finally, the statistical analysis of the comparisons is not described and is poor. Although ROC is probably the best analysis, the use of "accuracy" independently of the rest of the elements of the confusion matrix is not entirely appropriate. A more effective measure is that based on the Matthews Correlation Coefficient, which is much more illustrative of the behavior of the matrix, especially in the case of binary classifications.

R: Thank you for your remark about the efficiency of the Matthews Correlation Coefficient which seems particularly suitable for binary classification of unbalanced data as demonstrated in Chicco and Jurman, 2020. We included this measure in the revised version. A succinct statistical analysis using a Mann-Whitney U test was presented in Section 3.2.4 Summary of results.

Minor comments

3.1 ERP analysis:

- Fig 3 Where the baselines were taken?

R: Thank you for this remark. The baselines for Figure 3 were taken between -500 ms and 0 ms. To better clarify this point, we added the following sentence in Section 2.4 of the revised version: “The averaged signal value of the pre-stimulus interval from -500 ms to 0 ms was divided from the epochs for baseline correction”

- In the description of figure 4, in the main text, figures 4 A and B are cited instead of the left side and right side of figure 4.

R: Thank you for this comment, it is now modified in the revised version.

In the legend of figure 4 (left) it is said that the signals were recorded between 100 and 300 ms but it must be between 0 and 300 ms.

R: Thank you for this comment, it is now modified in the revised version.

3.2.1 Validation of the electrode selection:

Results about Fig. 6. Statistical analyzes are not mentioned.

In Fig. 6a, the authors talk about global maxima using 9 electrodes, but it is not stated how it is shown that there are significant differences between the electrode configurations. On the contrary, it seems that when more than 2 electrodes were used, the accuracy did not increase significantly.

Similarly for Fig. 6b, the description of the results lacked statistical analysis.

R: Thank you for your remark concerning the number of electrodes. Even though we report in Section 3.2.1 “The effect of the number and position of the electrodes” that “the classification score for the Tunnel vs. Checkerboard reaches a global maximum using 9 electrodes” , we used the whole set of 12 electrodes to train the main classification algorithms and report the main results of the article. The purpose of checking the classification scores with regards to the number and position of the electrodes was to evaluate if adding more electrodes would effectively add more information that would increase the classification scores. The fact that we use the whole set of 12 electrodes or 75 mean cortical sources activations was made clearer by adding the following sentences in Section 2.8: “xDAWN spatial filtering operating on the EEG signals from all 12 electrodes” and “xDAWN spatial filtering operating on the 75 averaged sources activations”. Moreover, we added the following paragraph in Section 3.2.1: “Considering that the positions of the 12 electrodes are uniformly distributed on the scalp and that the cross-validated classification accuracy reached a plateau before the 12th electrode, we can reasonably conclude that adding EEG signals from more electrodes will not substantially increase the amount of non-artifactual information”.

3.2.2 Inter-subject class…

- Fig. 7 C (and 8C and 9C) the description of the color code is mixing.

. … Accuracies are significantly higher… Where are the statistics??

R: Thank you for your comment. Regarding the color code, we now uniformized the color code across the new figures in the revised version as you suggested. 

Regarding the words “significantly higher”, it was indeed a poor choice of words since the expression could be misinterpreted as statistically significantly higher, thus your comment about the reporting of the statistics. Since, in the section about the impact of the number and position of electrodes, a statistical analysis of the results will not add meaningful information. Thus we chose to remove the word “significantly” where no statistical tests were not performed and reported. 

When the word “significantly” was used in the article, statistical tests were performed and reported in Section 3.2.4 Summary of results.

3.2.4 inter-subject and intra-subject …

- It is not clear in the text what is to be shown with figures 7C and 8C. Does this comparison refer to the gray images that were presented with the checkerboard and with the tunnels?

If so, what the results would be saying is that the classification system between the gray images is more related to the checkerboard and the tunnel than to the gray images themselves. This requires further explanation in the text.

R: Thank you this pertinent remark. We now better explained the fact that the two types of grey images were not exactly the same depending on the preceding checkerboard or tunnel image. In other words, the grey screens play the role of avoiding a post-fusion between checkerboard and tunnel but, and were thus influenced by the type of preceding visual stimulation. The ERP and RSTA analysis of grey images were move to the supplementary material section for better clarity. 

- Fig. 9. Must be Intra-subjects or Inter-subjects instead intra-patients or inter-patients

R: Thank you for this remark, we now changed this in the text and in the graphs of the revised version.

---

## [Decision Letter · Decision Letter 1]

27 Oct 2021

PONE-D-21-04037R1Riemannian classification of single-trial surface EEG and sources during checkerboard and navigational images in humansPLOS ONE

Dear Dr. Simar,

Thank you for submitting your manuscript to PLOS ONE. After careful consideration, we feel that it has merit but does not fully meet PLOS ONE’s publication criteria as it currently stands. Therefore, we invite you to submit a revised version of the manuscript that addresses the points raised during the review process. While one of the original reviewers approved publication, the reviewer that previously asked for making the material available as per Plos policy, provided now complete comments. This reviewer finds that the manuscript has been improved and is easier to follow, nevertheless, he/she still suggests a revision asking for additional analysis related to the classification method and interpretation of this approach. Please submit your revised manuscript by Dec 11 2021 11:59PM. If you will need more time than this to complete your revisions, please reply to this message or contact the journal office at plosone@plos.org. Please include the following items when submitting your revised manuscript:A rebuttal letter that responds to each point raised by the academic editor and reviewer(s). You should upload this letter as a separate file labeled 'Response to Reviewers'.A marked-up copy of your manuscript that highlights changes made to the original version. You should upload this as a separate file labeled 'Revised Manuscript with Track Changes'.An unmarked version of your revised paper without tracked changes. You should upload this as a separate file labeled 'Manuscript'.

We look forward to receiving your revised manuscript.

Kind regards,

Yury Ivanenko

Academic Editor

PLOS ONE

Reviewers' comments:

Reviewer's Responses to Questions

**Comments to the Author**

1. If the authors have adequately addressed your comments raised in a previous round of review and you feel that this manuscript is now acceptable for publication, you may indicate that here to bypass the “Comments to the Author” section, enter your conflict of interest statement in the “Confidential to Editor” section, and submit your "Accept" recommendation.

Reviewer #1: (No Response)

Reviewer #3: All comments have been addressed

2. Is the manuscript technically sound, and do the data support the conclusions?

Reviewer #1: Yes

Reviewer #3: Yes

3. Has the statistical analysis been performed appropriately and rigorously? 

Reviewer #1: Yes

Reviewer #3: Yes

4. Have the authors made all data underlying the findings in their manuscript fully available?

Reviewer #1: Yes

Reviewer #3: Yes

5. Is the manuscript presented in an intelligible fashion and written in standard English?

Reviewer #1: Yes

Reviewer #3: Yes

6. Review Comments to the Author

Reviewer #1: Thank for making the material available as per Plos policy. As for the manuscript, and given the reviewer 2 revision I think it now look great and easy to follow. Yet, I have a major problem with the manuscript. The paper shows, convincingly, that XDAWN+Riemannian projection at the geometric mean gives good results. The problem is that other classification pipelines not using XDAWN and not using Riemannian geometry can also discriminate between conditions. In my opinion, for a publication in a general outlet you need more than showing it works, you need a comparison. Not a comparison to a grand average, which make no sense since the framework (fitting vs predicting) are different, but a comparison of performances with the data without XDAWN or without projecting onto the tangent space. My point is you need to show it outperforms other approaches or at least provides a more straightforward interpretation.

Given the authors track record, I'm sure they are well aware of the various other methods out there. Just pointing out for instance in the introduction when it sounds like there is noting else but grand averages and single trial analyses are new ... well see eg https://www.frontiersin.org/articles/10.3389/fpsyg.2011.00322/full and references therein - starting with Donchin, E. (1969). Discriminant analysis in average evoked response studies: the study of single-trial data. Electroencephalogr. Clin. Neurophysiol. 27, 311–314.

Concretely, my suggestion is to a logistic regression on the data, then with XDAWN, then in tangent space without DAWN - that way we have all 4 ways to process and show your approach works best ; otherwise you just say that you can classify like many other can. I don't mean to sound harsh, I think the method is great and the paper is well written, but it's not convincing enough, especially in a generic outlet.

Dr Cyril Pernet

Reviewer #3: All comments on the manuscript have been reviewed. The manuscript has been considerably improved and is ready for publication in my opinion.

7. PLOS authors have the option to publish the peer review history of their article (what does this mean?). If published, this will include your full peer review and any attached files.

Reviewer #1: **Yes: **Cyril Pernet

Reviewer #3: No

---

## [Author Response · Author response to Decision Letter 1]

17 Dec 2021

Reviewer #1:

Reviewer’s comment:

Thank for making the material available as per Plos policy. As for the manuscript, and given the reviewer 2 revision I think it now look great and easy to follow. Yet, I have a major problem with the manuscript. The paper shows, convincingly, that XDAWN+Riemannian projection at the geometric mean gives good results. The problem is that other classification pipelines not using XDAWN and not using Riemannian geometry can also discriminate between conditions. In my opinion, for a publication in a general outlet you need more than showing it works, you need a comparison. Not a comparison to a grand average, which make no sense since the framework (fitting vs predicting) are different, but a comparison of performances with the data without XDAWN or without projecting onto the tangent space. My point is you need to show it outperforms other approaches or at least provides a more straightforward interpretation.

Given the authors track record, I'm sure they are well aware of the various other methods out there. Just pointing out for instance in the introduction when it sounds like there is noting else but grand averages and single trial analyses are new ... well see eg https://www.frontiersin.org/articles/10.3389/fpsyg.2011.00322/full and references therein - starting with Donchin, E. (1969). Discriminant analysis in average evoked response studies: the study of single-trial data. Electroencephalogr. Clin. Neurophysiol. 27, 311–314.

Concretely, my suggestion is to a logistic regression on the data, then with XDAWN, then in tangent space without DAWN - that way we have all 4 ways to process and show your approach works best ; otherwise you just say that you can classify like many other can. I don't mean to sound harsh, I think the method is great and the paper is well written, but it's not convincing enough, especially in a generic outlet.

Authors’ response:

Thank you for your important comment and suggestion. In the previous version of the manuscript, although we showed that our approach (xDAWN filtering and Riemannian projection) is able to discriminate between the presentation of the two displays, we did not indeed show a comparison of multiple pipelines and thus did not show that our approach outperformed other approaches. 

You are right that it is important to show the comparative advantage of our method with respect to other classification pipelines and this would be a considerable improvement of the manuscript, as well as important information for the readers. Therefore, in the new version of the manuscript, we added section “3.2.5 Comparison of classification pipelines” that describes and compares 5 additional classification pipelines with our approach in order to show the benefits of applying xDAWN filtering and Riemannian projection, as well as to establish the robustness and generalizability of our approach with respect to the other classification pipelines. As you suggested, the additional pipelines include the use of a logistic regression on the vectorized EEG signals or cortical sources, as well as the selective use of xDAWN filtering and Riemannian projection. We also added a classification pipeline using the CSP algorithm. The results of these comparisons are presented with statistical significance tests computed with a Mann–Whitney U test. 

These results show that 

• the classification pipeline used in our approach outperforms the other pipelines in both intra-subject and inter-subject conditions, using either EEG signals or cortical sources

• xDAWN spatial filtering significantly improves inter-subject generalizability

• the projection onto the tangent space significantly improves the classification accuracy in the intra-subject condition

• the classification pipeline used in our approach significantly outperforms the CSP algorithm in the inter-subject condition

Please find hereunder the complete section that we added in the new version of the manuscript.

Lastly, the reference you suggested, referring to the relevance of single-trial analysis for neuroimaging data, was interesting and we thus included this sentence in the new version of the manuscript: “The relevance of considering single-trial analyses with neuroimaging data is also discussed in [42].”

3.2.5 Comparison of classification pipelines

As suggested by a reviewer, after the comparative analysis in section 2.8, we compare here the classification accuracy of our approach with other classification pipelines using the same cross-validation methodology. The purposes of this comparison are to (i) quantify the benefits of applying xDAWN filtering and projecting covariance matrices onto the tangent space, and (ii) establish the robustness and generalizability of our approach with respect to other classification pipelines.

To that end, Figure 9 compares the mean cross-validated accuracy, with respect to the intra-subject or inter-subject conditions, computed on the following classification pipelines with and without inverse modelling:

• LR on vectorized EEG signals/sources activations: the EEG signals or cortical sources activations are unrolled into a 1-dimensional vector and classified using a logistic regression (LR).

• Covariance matrices and LR: covariance matrices of EEG signals or cortical sources are estimated, unrolled into a 1-dimensional vector and classified using a logistic regression.

• Covariance matrices, TS and LR: covariance matrices of EEG signals or cortical sources are estimated, projected onto the tangent space and classified using a logistic regression.

• xDAWN covariance and LR: covariance matrices are estimated from the result of xDAWN spatial filtering applied on EEG signals or cortical sources, unrolled into a 1-dimensional vector and classified using a logistic regression.

• CSP and LR: the output of the Common Spatial Patterns (CSP) algorithm [59], [60] applied on EEG signals or cortical sources is classified using a logistic regression.

• xDAWN covariance, TS and LR: covariance matrices are estimated from the result of xDAWN spatial filtering applied on EEG signals or cortical sources, projected onto the tangent space and classified using a logistic regression.

The comparison of classification results from these different pipelines illustrated in Figure 9 highlights several interesting patterns. Firstly, we can observe that the classification pipeline used in our approach outperforms the other pipelines in both intra-subject and inter-subject conditions, using either EEG signals or cortical sources. Secondly, we observe that the use of xDAWN spatial filtering significantly improves the classification accuracy in the inter-subject condition (p < 0.001), i.e. inter-subject generalizability, but not in the intra-subject condition (p > 0.001). Whereas, conversely, the use of the projection onto the tangent space significantly improves the classification accuracy in the intra-subject condition (p < 0.001) but not in the inter-subject condition (p > 0.001), although this latter absence of accuracy improvement in the inter-subject condition may also be explained by the difficulty to discriminate inter-subject EEG signals and cortical sources without xDAWN spatial filtering for this task. Lastly, we observe that the classification pipeline used in our approach significantly outperforms the CSP algorithm in the inter-subject condition (p < 0.001) but not in the intra-subject condition (p > 0.001). These statistical comparisons were computed using a Mann–Whitney U test.

<Please see the figure in the "Response to Reviewers.pdf" file attached>

Figure 9. Comparison of the mean accuracies reached by different inter-subject and intra-subject classification pipelines based on EEG signals and (sub)cortical sources. The asterisk designates the classification pipeline used in our approach.

Reviewer #3:

Reviewer’s comment:

All comments on the manuscript have been reviewed. The manuscript has been considerably improved and is ready for publication in my opinion.

Authors’ response:

Thank you for your previous comments that contributed to the considerable improvement of the manuscript and for accepting it for publication.

---

## [Decision Letter · Decision Letter 2]

24 Dec 2021

Riemannian classification of single-trial surface EEG and sources during checkerboard and navigational images in humans

PONE-D-21-04037R2

Dear Dr. Simar,

We’re pleased to inform you that your manuscript has been judged scientifically suitable for publication and will be formally accepted for publication once it meets all outstanding technical requirements.

Kind regards,

Yury Ivanenko

Academic Editor

PLOS ONE

Additional Editor Comments (optional):

Reviewers' comments:

Reviewer's Responses to Questions

**Comments to the Author**

1. If the authors have adequately addressed your comments raised in a previous round of review and you feel that this manuscript is now acceptable for publication, you may indicate that here to bypass the “Comments to the Author” section, enter your conflict of interest statement in the “Confidential to Editor” section, and submit your "Accept" recommendation.

Reviewer #1: All comments have been addressed

2. Is the manuscript technically sound, and do the data support the conclusions?

Reviewer #1: Yes

3. Has the statistical analysis been performed appropriately and rigorously? 

Reviewer #1: Yes

4. Have the authors made all data underlying the findings in their manuscript fully available?

Reviewer #1: Yes

5. Is the manuscript presented in an intelligible fashion and written in standard English?

Reviewer #1: Yes

6. Review Comments to the Author

Reviewer #1: (No Response)

7. PLOS authors have the option to publish the peer review history of their article (what does this mean?). If published, this will include your full peer review and any attached files.

Reviewer #1: **Yes: **Cyril Pernet

---

## [Editor Report · Acceptance letter]

5 Jan 2022

PONE-D-21-04037R2 

Riemannian classification of single-trial surface EEG and sources during checkerboard and navigational images in humans 

Dear Dr. Simar:

I'm pleased to inform you that your manuscript has been deemed suitable for publication in PLOS ONE. Congratulations! Your manuscript is now with our production department. 

Kind regards, 

on behalf of

Dr. Yury Ivanenko 

Academic Editor

PLOS ONE